# All-inorganic perovskite quantum dot light-emitting memories

Meng-Cheng Yen[1,5], Chia-Jung Lee[1,5], Kang-Hsiang Liu[1], Yi Peng[1], Junfu Leng[2], Tzu-Hsuan Chang[3], Chun-Chieh Chang [1✉], Kaoru Tamada [2,4✉] & Ya-Ju Lee [1✉]

Field-induced ionic motions in all-inorganic $CsPbBr_3$ perovskite quantum dots (QDs) strongly dictate not only their electro-optical characteristics but also the ultimate optoelectronic device performance. Here, we show that the functionality of a single $Ag/CsPbBr_3/ITO$ device can be actively switched on a sub-millisecond scale from a resistive random-access memory (RRAM) to a light-emitting electrochemical cell (LEC), or vice versa, by simply modulating its bias polarity. We then realize for the first time a fast, all-perovskite light-emitting memory (LEM) operating at 5 kHz by pairing such two identical devices in series, in which one functions as an RRAM to electrically read the encoded data while the other simultaneously as an LEC for a parallel, non-contact optical reading. We further show that the digital status of the LEM can be perceived in real time from its emission color. Our work opens up a completely new horizon for more advanced all-inorganic perovskite optoelectronic technologies.

[1] Institute of Electro-Optical Engineering, National Taiwan Normal University, Taipei, Taiwan. [2] Institute for Materials Chemistry and Engineering (IMCE), Kyushu University, Fukuoka, Japan. [3] Graduate Institute of Electronics Engineering, National Taiwan University, Taipei, Taiwan. [4] Advanced Institute for Materials Research (AIMR), Tohoku University, Sendai, Japan. [5] These authors contributed equally: Meng-Cheng Yen, Chia-Jung Lee. ✉email: chang48@ntnu.edu.tw; tamada@ms.ifoc.kyushu-u.ac.jp; yajulee@ntnu.edu.tw

Recently, the demand for the next-generation memory devices featuring high switching speed, large storage capacity, and low power consumption has considerably grown, as the internet of things (IoT) and artificial intelligence (AI) technologies continue to dominate for a vast variety of consumer products used in day-to-day life. To meet this ever-increasing requirement, various kinds of high-performance advanced memory devices have been eagerly pursued in the semiconductor industry. Among them, the nonvolatile resistive random-access memory (RRAM, in which the stored information does not vanish when the applied power is turned off) constructed by simple metal/insulator/metal thin-film stacks, is an essential building block for this ongoing digital revolution[1–3]. The current–voltage (I–V) characteristics of the RRAM exhibit an abrupt change in electrical resistance between a high-resistance state (HRS) and a low-resistance state (LRS). One possible mechanism causing such a disparity is attributed to the formation and annihilation of the conductive filament bridging the two metal electrodes under the variation of the applied electric field (including both its direction and magnitude) across the RRAM, which consequently leads to the desirable electric-field-driven resistive switching. The HRS and LRS of the RRAM, analogous to the basic Boolean true and false expressions, are then used to represent the logic "0" and "1" signals in the digital electronic circuits, respectively. As the readout process of these two logic signals in an RRAM relies on the resistance measurement in a series sequence triggered by an external bias, the overall transmitted data rate using RRAM is fundamentally limited. A modified RRAM scheme that allows and compiles the parallel reading processes is therefore highly anticipated.

Optically readable memories, which integrate a conventional RRAM with a light-emitting diode (LED), are recently developed to overcome this issue, in which the HRS and LRS of the RRAM are recognized directly by the absence or presence of electroluminescence (EL) emissions from the LED[4–8]. Such a combination of the RRAM and the LED brings about a new memory device dubbed light-emitting memory (LEM), which enables parallel and synchronous reading processes of encoded signals via both electrical and optical detections. Compared to the RRAM solely using the electronic reading, the additional optical reading of the LEM offers much larger flexibility to carry more information and to realize multilevel data storage[9–11]. Additionally, the noncontact optical readout and the high-speed optical transmission inherently in the LEM are expected to expand the functionality of the conventional RRAM, especially in the fields of computation and communication[12,13]. However, the previous LEMs proposed in the literature are usually produced by integrating two separate devices comprised of completely dissimilar material systems. Many restrictions and issues arise in such hybrid LEMs in terms of their manufacturing compatibility, fabrication simplicity, and transmission synchronization of electrical and optical signals. For instance, in Refs[6] and [7]., the LEM devices are formed by stacking a $SiO_2$-based memory on top of a GaN-based light emitting diode. These vertically integrated structures for LEM are fairly complicated, and careful considerations must be made to ensure the proper device function. In these LEM structures, both the memory material and the common electrode that connects the memory to the underlying light emitter have to be transparent to the optical emission. The common electrode (graphene sheet in these cases) needs to have a high electrical conductivity and more importantly to provide good current spreading. An interfacial layer of metal nanoparticles is also needed to enhance the ON/OFF ratio of the memory. All of these requirements not only pose a serious constraint on the material selection and compatibility, but significantly add the complexity in the device fabrication. Also, none

of these demonstrations shows the operation speed of their LEM devices, which makes it unclear whether these devices can indeed be deployed for the real-world applications. Therefore, there is an urgent need for a novel LEM structure based on one specific material facilitating seamless integration between the photonic system and the electrical resistive switching to overcome these challenges.

To date, all-inorganic metal halide perovskite materials, usually expressed as the chemical formula of $ABX_3$, where A and B are metal cations with different sizes and X being a halide anion, have been intensively investigated due to their unique photophysical characteristics and high stability in the ambient environment[14,15]. They can be easily processed by solution methods and be implemented in low-cost, mass-production of optoelectronic devices for high-scalable applications. Furthermore, all-inorganic perovskite halides have low activation energies for the migration of halide-ion vacancies under an applied bias, which enables the drifts of charged cations and anions per the poling direction and consequently, forms a reversible p-i-n homojunction structure[16,17]. These field-induced ionic motions are believed to be the cause of several intriguing phenomena observed in the perovskite optoelectronic devices, for example, the hysteresis in the I-V curves in the perovskite solar cell[18,19] and the ionization-doping enhanced EL emissions in the perovskite light-emitting electrochemical cell (LEC)[20–22]. There are also demonstrations showing that the high ionic conductivity associated with the migration of halide-ion vacancies leads to significant resistive switching in the perovskite RRAM[23–25]. By taking advantage of the fast, electrically switchable ionic motions and the corresponding bifunctionalities (that is, RRAM and LEC) in a single perovskite structure, here we propose and demonstrate a novel all-inorganic perovskite $CsPbBr_3$ LEM by monolithically integrating a perovskite LEC with a perovskite RRAM. We show that this all-perovskite LEM could electrically read the encoded data by its RRAM and optically transmit the information through the emission from its LEC at a speed of 5 kHz. We further demonstrate a two-color emitting LEM by employing QDs with two different sizes in the device, enabling the real-time reading of the LEM digital status (either write or erase) simply by its emission colors. All of these new functionalities of our all-perovskite LEM demonstrated in this work are unprecedented, and cannot be achieved neither in the aforementioned vertically integrated LEM employing two dissimilar material systems[6,7], nor the single silicon nitride-based[8] or polymer bistable[4,5] light-emitting device. We propose a physical picture outlining the movements of each ion in the perovskite LEM and their reduction and oxidation processes under different bias scenarios by carefully investigating its intriguing electronic and optical characteristics. The demonstrated all-perovskite LEM promises not only a much higher integration capability than the hybrid ones previously reported, but also a much faster data processing and larger storage space. This work could also serve as a new paradigm for generating more advanced all-inorganic perovskite optoelectronics and new applications through the synergy of electronics and photonics.

## Results

**$CsPbBr_3$ QD-based LEM device structure and QD properties.** Figure 1a schematically presents the configuration of our all-inorganic perovskites-based LEM (top plot), composed of two nominally identical devices with a dimension of ~ 3.0 mm × 3.0 mm. Each has dual functionalities either as RRAM or LEC, completely switchable by manipulating the electric field direction across the LEM (bottom plot, Fig. 1a), as can be seen later in the article. A set of two devices is fabricated on top of a glass substrate with 250-nm-thick indium tin oxide (ITO) rectangular pads

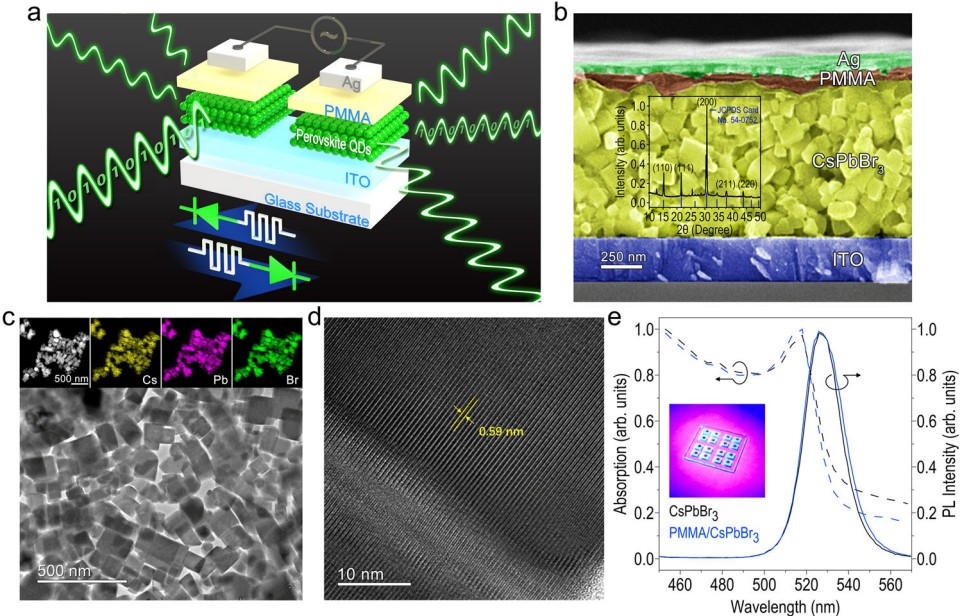

**Fig. 1 CsPbBr₃ QD-based LEM device structure and QD properties. a** Schematic of the CsPbBr₃ QD-based LEM device composed of two nominally identical devices (top plot), with equivalent electronic circuit symbols illustrating that each device has dual functionalities either as RRAM or LEC, depending on the electric field direction across the LEM (bottom plot). **b** Cross-sectional scanning electron microscope (SEM) image of the as-fabricated LEM, where the scale bar is 250 nm. XRD pattern of the CsPbBr₃ QDs along with that of the cubic structure (JSPDS No. 54-0752) is inserted in the figure. **c** TEM image of the CsPbBr₃ QDs. Insets show the EDS mapping of the CsPbBr₃ QDs for analyzing the elements of Cs, Pb, and Br. **d** High resolution TEM of the CsPbBr₃ QDs to identify the interplanar space of (200) plane. A clear fused interface is also observed in this figure. **e** Absorption (primary vertical axis) and PL spectra (secondary vertical axis) of the CsPbBr₃ QD films with and without the PMMA protection layer. Inset: A photograph of the as-fabricated LEM taken under the illumination of 405 nm UV-LED.

(4.0 mm × 7.0 mm), formed by RF magnetron sputtering through a shadow mask. Although the spatial distance between two devices is ~1.0 mm, they are connected electrically in series as the ITO has a low resistivity of ~ $4.8 \times 10^{-4}$ Ω·cm. Then, close-packed CsPbBr₃ QDs synthesized by the supersaturated recrystallization (SR) method at room temperature (see Methods) are spun coated on top of the ITO pads at 3000 rpm for 30 s, forming a quasi-continuous film with a thickness of ~800 nm for the LEM device active layer. Here the CsPbBr₃ QDs are preferred than bulk perovskites for two reasons. First, in order to form the latter, either the perovskite precursor solution[26–28] or the mixed powers of PbBr₂ and CsBr[29–31] would need to be subject to a period of high temperature for facilitating the nucleation and crystallization of the bulk perovskites, but the heating process would introduce undesirable thermal stress to the underlying ITO and reduce the electrical conductivity of ITO[32,33] and degrade the overall LEM device performance (Supplementary Fig. 1). Moreover, the perovskites QDs would offer a great opportunity compared to its bulk counterpart utilizing their inherent quantum confinement effect to significantly enhance the LEM device functionalities (Supplementary Fig. 2). A polymethyl methacrylate (PMMA) protection layer is then coated on top of the CsPbBr₃ QDs at the same spin speed. Finally, a 100-nm-thick circle- or square-patterned (~1.0 mm in diameter or in side length) silver (Ag) electrode, usually implemented for increasing the ON/OFF ratio of the RRAM[34], is deposited on top of the PMMA as contact electrodes by RF magnetron sputtering. Sharp boundaries between ITO/CsPbBr₃ QDs and PMMA/CsPbBr₃ QDs layers are well distinguished and clearly observed (Fig. 1b). The X-ray diffraction (XRD) pattern of the CsPbBr₃ QDs (inserted in Fig. 1b) exhibits a monoclinic crystal phase, which is similar to that of the cubic structure (JSPDS No. 54–0752) and can be identified by the split of the diffraction peak at ≈31° corresponding to (200) plane. This indicates the good crystallographic quality

and successful synthesis of the CsPbBr₃ QDs. The crystalline structure and the arrangement of the atomic plane of the CsPbBr₃ QDs are further examined by using transmission electron macroscope (TEM). The elemental composition of the CsPbBr₃ QDs is also analyzed by using the energy dispersive X-ray spectroscopy (EDS), as seen in Fig. 1c insets. The EDS mapping clearly illustrates that the Cs, Pb, and Br are homogeneously distributed inside the CsPbBr₃ QDs with the ratio of elemental amount as 20 (Cs): 15 (Pb) :65 (Br). The as-synthesized CsPbBr₃ QDs tend to aggregate with each other (Fig. 1c), and clear fused interface can be observed between two adjacent QDs (Fig. 1d). These results suggest the transfer of Cs⁺, Pb²⁺, and Br⁻ ions from the soluble to insoluble solvents is too fast during the supersaturated recrystallization process, which significantly enlarges the QD size to ~50–100 nm, larger than the CsPbBr₃ QDs synthesized by the hot-injection method that generally have a smaller size of ~ 10–15 nm[14,15]. Nevertheless, the well-resolved lattice fringes with an average interplanar space of 0.59 nm can be observed (Fig. 1d), consistent with the distance between the (200) planes indexed in the XRD measurement. Notably, the dimensions of CsPbBr₃ QDs with high crystallization are similar to the grain size of the perovskite films produced by the precursor solution with post-annealing treatment[26–28], suggesting that the CsPbBr₃ QDs are suitable not only for the LEM device active layer, but also for many other perovskite optoelectronic devices[35,36]. A sharp emission peak centered at λ = 526 nm with a comparable photo-luminescence (PL) intensity is observed in both the CsPbBr₃ QDs and PMMA/CsPbBr₃ QDs samples (Fig. 1e). A photograph of the fabricated LEM devices taken under the illumination of 405 nm UV-LED is also inserted in the figure. The same amount of Stokes shift of ~8 nm is determined for both CsPbBr₃ QDs samples with and without protective PMMA as they have a similar excitonic absorption peak located at λ = 518 nm. The PMMA protection layer in the structure provides an effective way

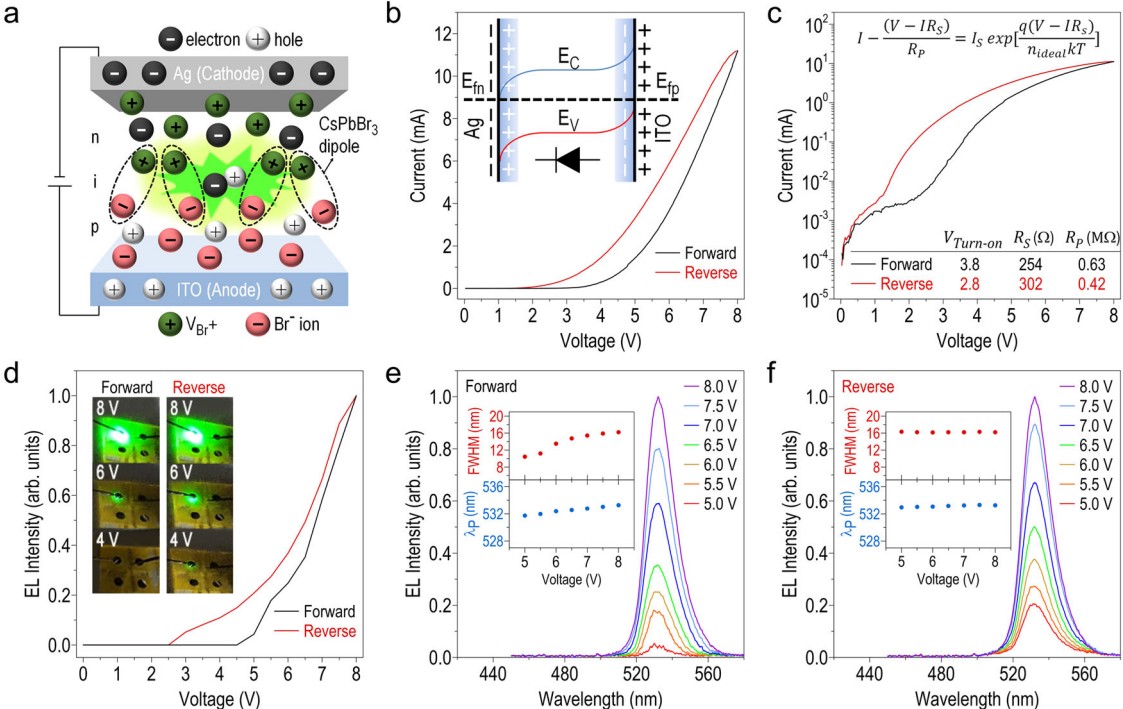

**Fig. 2 Electro-optical characteristics of CsPbBr₃ QD-based LEC. a** Schematic illustration of ion migration inducing electric dipoles aligning along the direction of an applied electric field. Cations and anions are hence accumulated close to the Ag and ITO electrodes, respectively, leading to the formation of the p-i-n diode. **b** I–V characteristics of the perovskite LEC for the sweep voltage range of 0 V → 8 V → 0 V, and a clear hysteresis effect is observed in this figure. Inset: the corresponding energy-level diagram while applying a negative poling voltage on the LEC (Ag: cathode; ITO: anode). **c** Same as **b**, but plotted in a semi-log scale. Insets: the modified Shockley equation (top) and a summary table of the turn-on voltage, series and shunt resistances extracted from the forward and reverse I-V characteristics (bottom). **d** EL intensity of the perovskite LEC versus sweeping voltage in a range of 0 V → 8 V → 0 V. Insets show the images of the LEC under forward (left) and reverse (right) voltages of 4, 6, and 8 V. **e** and **f** Evolutions of EL spectra with the increase of forward and reverse voltages. The peak wavelength (λ_P) and the spectral bandwidth (FWHM) extracted from the EL spectra as a function of voltage are also inserted in both figures.

to prevent the Ag penetration during the sputtering deposition, as evidenced by Supplementary Fig. 3, which in turn minimizes the formation of current leakage paths between the top (Ag) and the bottom (ITO) electrodes, and enhances the performance and the stability of the fabricated LEM. The surface morphologies of the CsPbBr₃ QDs, the PMMA/CsPbBr₃ QDs, and the Ag/PMMA/CsPbBr₃ QDs films are also examined by the atomic force microscope (Supplementary Fig. 4).

**Electro-optical characteristics of CsPbBr₃ QD-based LEC.** We first investigate the emission characteristics of the LEM by applying a negative poling voltage to one of its constituent devices through the top Ag (cathode) and the bottom ITO (anode) electrodes. Since the perovskite materials are conceived as a class of solid electrolytes with good ionic conductivity, electric dipoles pointing along the direction of the electric field (that is, the bottom-up direction) can be induced inside the active layer, as plotted in Fig. 2a. Besides, the cations (i.e., Cs⁺ or bromide ion vacancy, $V_{Br}^+$) and anions (i.e., Br⁻) are further dissociated from the CsPbBr₃ QDs and which would drift, respectively, towards the Ag cathode and the ITO anode and accumulate there. The band bending of the CsPbBr₃ QDs at the interfaces is hence changed accordingly and leads to the formation of p-i-n diode (inset, Fig. 2b), which facilitates the photon emission through the radiative recombination of electron-hole pairs, and that is the working principle of the CsPbBr₃ QD-based LEC. Here shall be addressed that light emission can only be observed by applying the negative poling voltage on the LEC, while no light is emitted if changing the voltage polarity to positive. In addition, similar to the method for characterizing most of the

organic LEDs, the light emissions from our LEC are measured by a photodiode placed underneath the glass substrate, to ensure that the reflected light by the top Ag electrode within the device can also be collected. Figure 2b shows the typical I–V characteristics of the LEC under the negative poling voltage sweeping from 0 to 8 V (black curve, forward I–V) and then sweeping back from 8 to 0 V (red curve, reverse I–V). Both forward and reverse I–V characteristics exhibit well rectifying diode-like behaviors, suggesting a proper formation of p-i-n homojunctions[37]. Also, a clear electrical hysteresis effect is observed in this figure, mainly because the cations and anions accumulated at the cathode and anode cannot respond instantaneously to the applied voltage change. Note that no pre-poling voltage is applied to the LEC before acquiring its I–V characteristic (Fig. 2b), and the hysteresis effect is observed upon biasing the LEC and is repeatable after multiple sweeps. In Fig. 2c, we replot I–V characteristics of the LEC in a semi-log scale to identify several physical parameters by the modified Shockley equation expressed as[38]:

$$I - [(V - IR_S)/R_P] = I_S \exp[q(V - IR_S)/n_{ideal}kT] \quad (1)$$

where $R_S$, $R_P$, $n_{ideal}$, and $I_S$ are the series resistance, shunt resistance, ideality factor, and saturation current, respectively. The turn-on voltage, series and shunt resistances extracted from the forward and reverse I–V characteristics are $V_{turn-on}$ = 3.8 V, $R_S$ = 254 Ω and $R_P$ = 0.63 MΩ, and $V_{turn-on}$ = 2.8 V, $R_S$ = 302 Ω and $R_P$ = 0.42 MΩ, respectively. Compared to the forward I–V characteristic, the reverse one has smaller turn-on voltage and a little larger series resistance (but similar shunt resistance), and hence a stronger light intensity is expected during the revere sweeping process. Figure 2d

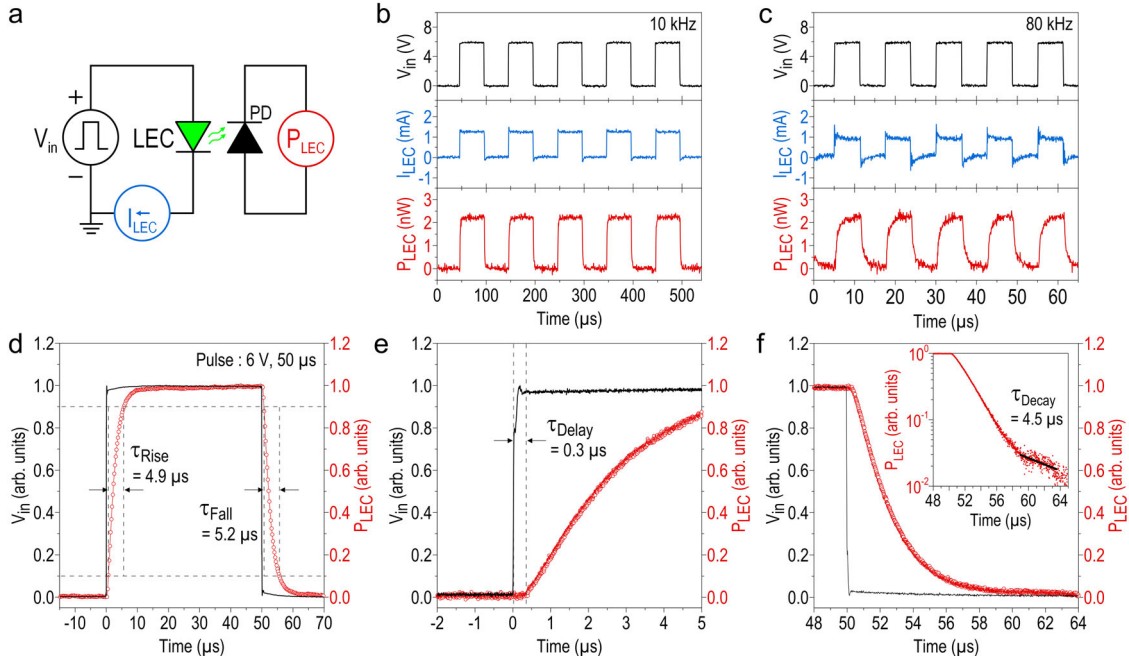

**Fig. 3 Transient response of CsPbBr₃ QD-based LEC. a** Circuit diagram illustrating the transient measurement setup for simultaneously monitoring the pulse voltage ($V_{in}$) applied on the LEC, the electrical current ($I_{LEC}$) flowing through the LEC, and the optical output-power ($P_{LEC}$) emitting by the LEC. **b, c** Time traces of $I_{LEC}$ and $P_{LEC}$ of the LEC under modulated $V_{in}$ at 10 kHz and 80 kHz. **d** Temporal $P_{LEC}$ in response to a square-wave pulse voltage of $V_{in}$ (magnitude: 6 V, pulse width: 50 μs, and duty cycle: 50%) applied on the LEC. The rise ($\tau_{Rise}$) and fall ($\tau_{Fall}$) times of $P_{LEC}$ are also marked in the figure. **e** The enlarged rising edge of $P_{LEC}$, where $\tau_{Delay}$, the time delay from the turn-on of $V_{in}$ till the onset of $P_{LEC}$, is estimated to be ~0.3 μs. **f** The enlarged falling edge of $P_{LEC}$ with the inset showing the same falling edge but plotted in a semi-logarithmical scale to extract the decay time ($\tau_{Decay}$) of the de-trapped carriers.

shows the EL intensity as a function of applied voltage for both forward (black curve) and reverse (red curve) sweeping processes. As expected, an optical hysteresis phenomenon is clearly observed due to the better reverse I-V characteristic of the LEC. The observed electrical and optical hysteresis phenomena do not significantly affect the interband optical transitions in the CsPbBr₃ QDs, as both the peak wavelength ($\lambda_P$ ~ 533 nm) and the spectral bandwidth (FWHM ~16 nm) obtained by the evolution of EL spectra are quite stable regardless of the applied voltage (Fig. 2e, f). It suggests that the LEC is a reliable light source for the parallel reading process and is suitable to be integrated in the LEM structure.

**Transient response of CsPbBr₃ QD-based LEC.** In addition to efficient EL emissions, a fast response of an LEC is also essential for applications in data communication and signal modulation. Figure 3a schematically depicts the setup for the transient measurement of our CsPbBr₃ QD-based LEC. The light output-power ($P_{LEC}$) of the LEC modulated by a pulse bias ($V_{in}$) is detected by a photodiode (PD), and the current ($I_{LEC}$) flowing through the LEC is analyzed over a series resistor. At the modulation frequency of 10 kHz (Fig. 3b), $P_{LEC}$ and $I_{LEC}$ fully follow the pulse bias applied to the LEC, and both of them are distorted to some extent (especially for $P_{LEC}$, see Fig. 3c) at a higher modulation frequency of 80 kHz (approaching to the transmission speed regulated for the near-field communication, that is, ~ a hundred of kbit/s). Nevertheless, $P_{LEC}$ is sufficiently saturated and thus can still be read to identify the on/off states of the LEC. Figure 3d shows temporal response of $P_{LEC}$ in response to a square-wave pulse bias of $V_{in}$ (magnitude: 6 V, pulse width: 50 μs, and duty cycle: 50%), where the response time is defined as the time difference between 10 and 90% of the maximum $P_{LEC}$. The extracted rise and fall times of $P_{LEC}$ are $\tau_{Rise}$ ~ 4.9 and $\tau_{Fall}$ ~ 5.2 μs, respectively. Both values are on a microsecond time scale, indicating that the LEC's

light emission responds instantly to the pulse bias, and more importantly the parallel data transmission using LEM is indeed feasible. Figure 3e plots the enlarged rising edge of $P_{LEC}$ to further examine its onset time $\tau_{Delay}$, the time delay from the initial rise of $V_{in}$ to the onset of $P_{LEC}$. A $\tau_{Delay}$ value of ~ 0.3 μs is extracted, representing the time scale over which the radiative recombination occurs in the perovskites when driven by $V_{in}$. Given the charge-carrier mobility ($\mu_{eff}$) is expressed as $\mu_{eff} = L/(\tau_{Delay} \cdot E)^{39}$, where L is the thickness of the active layer and E is the electric field across the LEC, we estimate $\mu_{eff}$ of the CsPbBr₃ QDs to be ~ $1.38 \times 10^{-3}$ cm² · V⁻¹ · s⁻¹ from the transient $P_{LEC}$ data. The smaller $\mu_{eff}$ value as compared to that of bulk perovskites[40–42] is expected and results from more scattering boundaries within the QD layer than in the bulk. Upon $V_{in}$ is turned off, the enlarged falling edge of $P_{LEC}$ in Fig. 3f shows that $P_{LEC}$ decays slowly to 3 % in ~ 8 μs, followed by a faster decay with a $\tau_{Decay}$ value of ~ 4.5 μs (inset, Fig. 3f). The two steps in the falling of $P_{LEC}$ have been observed elsewhere, resulting from the depletion of the carrier reservoir established under pulse bias[43], and the radiative recombination of delocalized carriers originally trapped by the defect centers within the perovskites[44], respectively. We estimate the trap center density to be $1.2 \times 10^{15}$ cm⁻³ by integrating $P_{LEC}$ over the entire period of $\tau_{Decay}$, which is in line with other highly-efficient perovskites reported in the literature[45,46].

**Resistive switching of CsPbBr₃ QD-based RRAM.** We then perform the resistive switching characteristics of the CsPbBr₃ QD-based RRAM by applying a positive poling voltage, as shown in Fig. 4a. In contrast to the LEC, here the anode is the top Ag and the cathode corresponds to the bottom ITO electrode. Ag, as the active metal, is easily oxidized and becomes Ag⁺ cation, as expressed by Ag → Ag⁺ + e⁻ (left panel, Fig. 4a). The constant

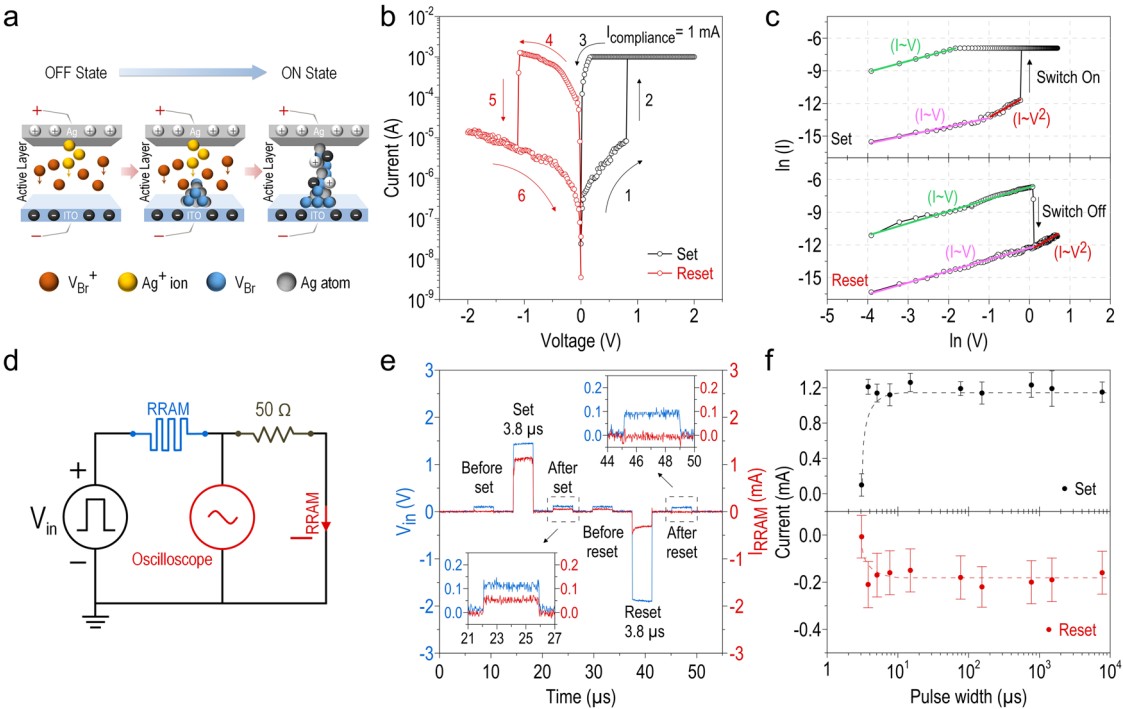

**Fig. 4 Resistive switching of CsPbBr₃ QD-based RRAM. a** Illustration of resistive switching in the CsPbBr$_3$ QD-based RRAM at the initial stage (left), during the migration of Ag$^+$ cations and V$_{Br}^+$ vacancies (middle), and the final formation of the V$_{Br}$ conducting channel and the Ag filament (right). **b** Typical *I–V* characteristic of the CsPbBr$_3$ QD-based RRAM by setting the compliance current to I$_{CC}$ = 1 mA. **c** Same as **b**, but plotted in a log-log scale with fitting lines corresponding to different conduction mechanisms (pink and red lines represent the linear Ohmic and the quadratic space-charge-limited conductions in HRS, respectively, whereas green line represents the linear Ohmic conduction in LRS) for Set (top) and Reset (bottom) conditions. **d** Circuit diagram of the measurement system for examining the resistive switching speed of the RRAM. **e** Transient responses of the RRAM during the set (1.5 V, 3.8 μs) and reset (−2.0 V, 3.8 μs) processes. Insets: enlarged time trace of the read pulses (0.1 V, 3.8 μs) fed into the RRAM after set (lower left) and reset (upper right) pulse biases. **f** I$_{RRAM}$ vs. pulse width of the applied bias V$_{in}$ for both set (top) and reset (bottom) processes. Error bars represent plus/minus standard deviation.

positive poling voltage further drifts Ag$^+$ cations to migrate over the entire CsPbBr$_3$ QDs active layer, and eventually reach to the bottom ITO cathode. Ag$^+$ cations are reduced back to neutral Ag atoms there, which serves as nucleation seeds for the subsequent growth of Ag clusters through repeated reduction processes of Ag$^+$ cations (middle panel, Fig. 4a). The Ag filament is hence produced inside the active layer and vertically connects the top and the bottom electrodes (right panel, Fig. 4a). A similar phenomenon has occurred for the bromide vacancy (V$_{Br}^+$), as it has been reported that most of the halide anions have low activation energy[17,47,48], leading to facile migrations for both bromide anions (Br$^-$) and bromide vacancy in the CsPbBr$_3$ QDs. When a positive poling voltage is applied to the RRAM, V$_{Br}^+$ is generated and then impelled towards the bottom ITO cathode. After being reduced there (as expressed by V$_{Br}^+$ + e$^-$ → V$_{Br}$), conducting channels are induced in a similar way as the formation of Ag filament. The processes for the formation of the V$_{Br}$ conducting channel and the Ag filament are reversible and responsible for the resistive switching effects in the RRAM. The ion migrations have been studied by using the SEM measurement and EDS analysis (Supplementary Fig. 5), and the results shall be applicable for elucidating the dynamic ionic transport and conduction processes of the LEM, as will be discussed in more detail below. Figure 4b shows the typical I-V characteristic of the as-fabricated RRAM by setting the compliance current to I$_{CC}$ = 1 mA. The RRAM exhibits a high electrical resistance of ~ 1.6 × 10$^5$ Ω under the reading bias of 0.50 V, and that represents the OFF state (or the HRS). During the positive voltage sweeping process from 0 to 2.0 and back to 0 V (or referred to as the SET process), the RRAM

readily switches to the ON state (or the LRS) at ~ 0.82 V with a low electrical resistance down to ~ 8.3 × 10$^2$ Ω (constrained by I$_{CC}$), equivalent to two orders of magnitude of resistance plummeting. The device then switches back to the OFF state at ~ −1.12 V while subject to a negative voltage sweeping process (from 0 to −2.0 to 0 V), which is termed as the RESET process. In Fig. 4c, we replot the I-V characteristic of the RRAM in a log-log scale [i.e., In (I)vs. ln(V)] under both positive (top) and negative (bottom) voltage sweeping processes to elucidate the possible mechanism for the conduction behavior of the OFF and the ON states. The HRS for both sweeping modes can be interpreted by the Ohm's law (I ~ V) and the space-charge-limited current (SCLC, also named as the Child law in which I ~ V$^2$) model in two separate voltage regions. The observed transition from the Ohmic behavior to the SCLC indicates that the unfilled trap centers (~1.2 × 10$^{15}$ cm$^{-3}$) are seized gradually by the charged carriers, leading to a larger current increment with the increased voltage. On the other hand, the conduction behavior of the LRS for both sweeping modes is well described by the Ohmic conduction accompanied by a dramatic reduction of the electrical resistance, indicating the V$_{Br}$ conducting channel and the Ag filament are well established inside the active layer in this stage. Note that the intrinsic oxygen-vacancy-rich nature inherited in the ITO electrode barely affects the resistive switching characteristics of the CsPbBr$_3$ QD-based RRAM, as illustrated in Supplementary Fig. 6.

For practical applications, a fast switching speed of an RRAM is highly desirable. We characterize the switching speed of our CsPbBr$_3$ QD-based RRAM using a measurement setup (see Fig. 4d) composed of a pulse generator, an oscilloscope, and a

load resistor (50 Ω). The transient switching between the HRS and the LRS, and the corresponding real-time current ($I_{RRAM}$) are directly monitored by examining the bias change over the load resistor. As shown in Fig. 4e, read pulses (0.1 V, 3.8 µs) are fed into the RRAM before and after the set/reset pulses. The RRAM is switched from the HRS to the LRS when a set pulse (1.5 V, 3.8 µs) is applied, evidenced by the surge of $I_{RRAM}$ in response to the read pulse (inset, lower left). The RRAM is then switched back to the HRS upon applying a reset pulse (− 2.0 V, 3.8 µs), and therefore, almost no current is detected by the read pulse (inset, upper right). This fast switching behavior of our RRAM holds even when the pulse widths of the set (1.5 V)/reset (− 2.0 V) bias are narrowed down to ~3.0 µs, as can be seen in Fig. 4f. As far as the device reliability is concerned, both the HRS and the LRS (read at 0.1 V) of the RRAM can retain for over $10^5$ sec and the ON/OFF current ratio keeps at ~$10^2$ (Supplementary Fig. 7a), indicating good retention stability. We also conduct an endurance test on our RRAM under a pulse sweeping mode (Supplementary Fig. 7b), and except for some fluctuations, the HRS/LRS ratio remains basically the same even after more than $10^4$ sweeping cycles. Moreover, the RRAM can be switched reversibly between the HRS and the LRS more than 50 times when subject to repeated sweeping processes between a positive and a negative dc bias (Supplementary Fig. 7c), and its $I$–$V$ characteristic shows a small device-to-device variation (Supplementary Fig. 8). All of these validate the reliable and reproducible write/erase characteristics of our CsPbBr$_3$ QD-based RRAM.

**Active switching of Ag/CsPbBr$_3$ QDs/ITO device functionalities**. Till now, we have demonstrated that a single CsPbBr$_3$ QD-based device can operate at a fast speed, either as an efficient LEC or a reliable RRAM, depending on its bias polarity. Real-time switching the device operation between the LEC and the RRAM modes by simply changing its bias polarity would be very attractive, and strongly impact the ultimate performance of an LEM by pairing such two devices, as will be discussed in more detail later. Here we explore the dual functionality of a single CsPbBr$_3$ QD-based device and its switching behavior by applying both dc and ac bias voltages, as illustrated schematically in Fig. 5a. As can be seen in the dc I-V characteristic in Fig. 5b, the CsPbBr$_3$ QD-based device acts as a typical RRAM with a rapid current increase at ~ 0.7 V (i.e., the set voltage) in the positive voltage sweep from 0 to 2 V, and the RRAM mode is switched off at −1.1 V (i.e., the reset voltage) in the negative voltage sweep. (Note that the slight discrepancy in the set voltage between Fig. 4b and here stems from the small device to device variation mentioned earlier.) Interestingly, under an even higher negative bias voltage the device is turned into an LEC, and when the negative voltage is increased to −3.2 V, bright EL emissions from the CsPbBr$_3$ QDs, are observed by naked eyes. An electrical hysteresis is clearly observed when reversing the negative voltage from −4 back to −1.1 V, beyond which the device is transformed back to a RRAM, proving the agile switching of the device functionality by changing the bias polarity. To characterize the switching speed, we then apply an alternating forward (+2 V) and reverse (−6 V) pulse voltage (pulse width = 0.25 ms) to the device. The temporal traces plotted in Fig. 5c show that fast switching between the LEC and the RRAM modes at a speed of 1 kHz is achieved. To the best of our knowledge, this frequency-response data represents the first experimental demonstration of actively switching a single CsPbBr$_3$ QD-based device between its two fundamentally different functionalities (i.e., light emission and data storage). We further characterize the frequency dependence of the $I_{RRAM}$ (Fig. 5d, top panel) and the $P_{LEC}$ (Fig. 5d, bottom panel), and the

measured data (solid circles) show that a 3-dB bandwidth (i.e., the frequency at which the signal drops to the half of its initial value) of ~ 1 kHz can be inferred. We expect that the switching cut-off frequency could be potentially enhanced by properly programming the input bias applied to the device, for example by using a combination of both ac and dc biases to mitigate the response time required for the ion transport. Nevertheless, we have experimentally verified for the first time that a CsPbBr$_3$ device can be electrically switched between the LEC and the RRAM modes on a sub-millisecond time scale. It will be discussed later that the swift switching between two functionalities in a single CsPbBr$_3$ device is essential for implementing an all-perovskite LEM when pairing such two identical devices, where the encoded information can be read by one CsPbBr$_3$ device acting as an RRAM and at the same time transmitted to the distant through the light emission from the other as an LEC.

**Electro-optical characteristics of CsPbBr$_3$ QD-based LEM and corresponding ionic migration processes**. We then series connect two nominally identical CsPbBr$_3$ QD-based devices to form an all-perovskite LEM and investigate its characteristics under a dc bias voltage. The bias voltage is applied to the top Ag electrode of the left device while that of the right device is grounded, so that the current flows from the left (right) to the right (left) during a positive (negative) voltage sweep. Figure 6a shows the variation of the EL intensity (top panel) and the $I$–$V$ characteristics (bottom panel) of the LEM when subject to a complete scanning cycle that includes both positive (0 V → +8 V → 0 V) and negative (0 V → −8 V → 0 V) voltage sweeps. Similar to the typical RRAM, an initial electroforming process is needed to conduct the LEM for achieving the resistive switching effect (Supplementary Fig. 9). The electroforming voltage required for switching the LEM from HRS to LRS is about −4.5 V, and a two-step hopping in the measured current is observed during the electroforming process, implying two different types of conducting channels induced in the left and right device of the LEM. During the positive voltage sweep, the EL emission is observed only from the right device of the LEM when the applied voltage exceeds ~ 5 V. The EL emission appears only from the left device of the LEM upon switching to a negative voltage sweep, as seen by the photographs inserted in the top panel of Fig. 6a. Clear optical hysteresis is again observed during both positive and negative voltage sweeps, due to the non-instantaneous response of the migrated ions to the applied voltage change as mentioned before. The distinct behaviors of the I-V characteristics of the LEM observed in four different positive bias regions (denoted as (I), (II), (III), and (IV), respectively; see bottom panel of Fig. 6a) represent four different dynamic ionic transport and conduction processes: (I) 0–0.9 V, where initial formation/reconstruction of the Ag and V$_{Br}$ filaments in the left device occur, (II) 0.9–1.9 V, in which two ON-state RRAM devices are connected in series, (III) 1.9–4.6 V, annihilation of the Ag filament and the rebuilding of V$_{Br}$ filament in the right device, and (IV) 4.6–8.0 V, the ON-state RRAM on the left is connected in series with the emitting LEC on the right. Possible scenarios of ion migrations in each bias region are plotted in Fig. 6b, with the corresponding band diagrams of the LEM operating in these regions shown in Supplementary Fig. 10. After the electroforming process, both the Ag filament and V$_{Br}$ conducting channel are well established in the right device, whereas only the V$_{Br}$ conducting channel is induced on the left (Supplementary Fig. 9b). A positive voltage in the range of 0–0.9 V erases the original V$_{Br}$ conducting channel of the left device through the oxidation process (V$_{Br}$ → V$_{Br}^+$ + e$^-$), without affecting the high conductivity of the right. The continuous supplement of positive voltage facilitates the gradual formation of

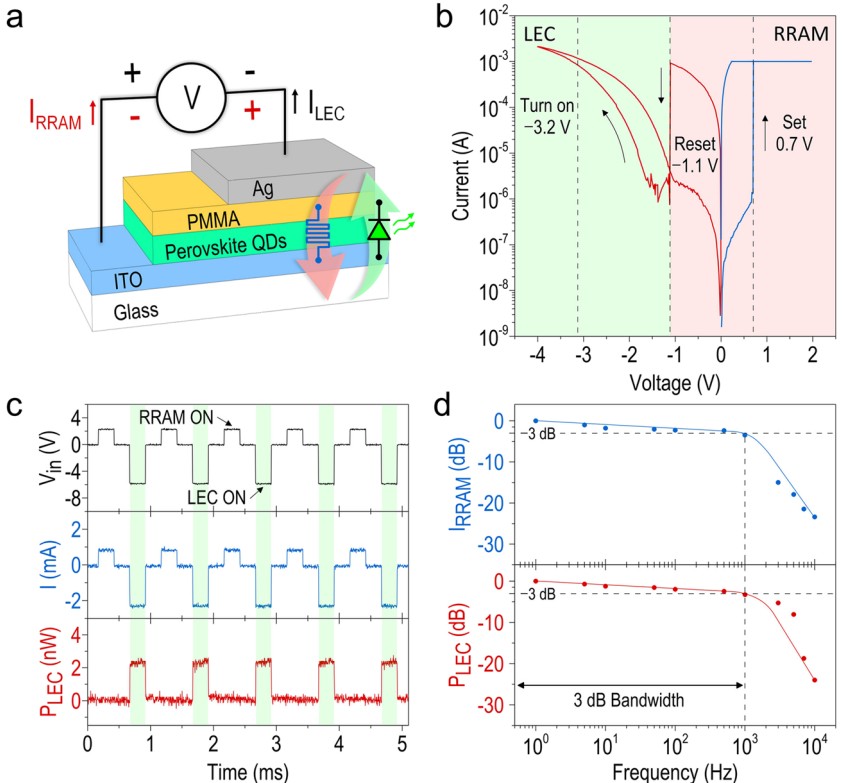

**Fig. 5 Active switching of Ag/CsPbBr₃ QDs/ITO device functionalities. a** Schematic illustrating the dual functionality of the CsPbBr₃ QD-based device either as RRAM or LEC by alternating the bias polarity. **b** I–V characteristic of the CsPbBr₃ QD-based RRAM under a dc bias sweep (0 V → +2 V → −4 V → 0 V). The set (0.7 V) and reset (−1.1 V) voltages of the RRAM mode, and the turn-on (−3.2 V) voltage of the LEC mode are also marked in the figure. **c** Demonstration of the functionality switching between the LEC and the RRAM modes in one device by using alternating forward and reverse bias of +2/−6 V with a 0.25 ms pulse width. **d** $I_{RRAM}$ (top panel) and $P_{LEC}$ (bottom panel) vs. operating frequency of the applied bias $V_{in}$. 3 dB bandwidth is indicated in the figure.

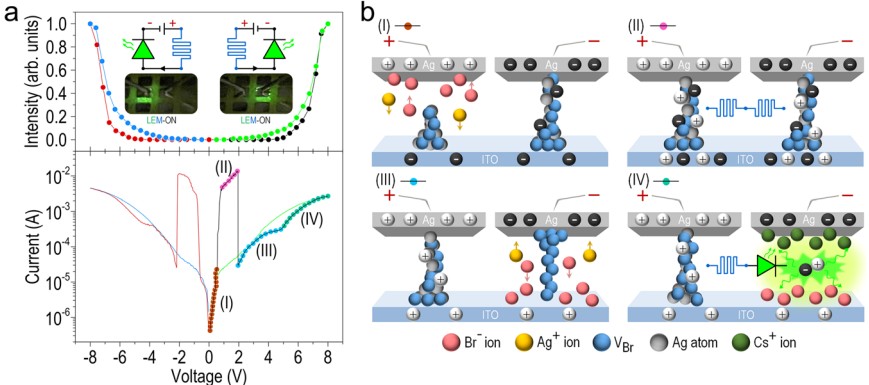

**Fig. 6 Electro-optical characteristics of CsPbBr3 QD-based LEM and corresponding ionic migration processes. a** Variations of EL intensity (top) and I-V characteristic (bottom) of the LEM when subject to a complete scanning cycle including both positive (0 V → +8 V → 0 V) and negative (0 V → −8 V → 0 V) voltage sweeps. The positive part of the I-V characteristic is divided into four regions to clarify the underlying mechanism for the observed optoelectronic characteristics of the CsPbBr₃ QD-based LEM. Insets: photographs taken while applying 8 V (right) and −8 V (left) to the LEM, showing the capability of the LEM for parallel optical and electrical readings. **b** Schematics illustrating ion migrations in the region of (I)–(IV).

the Ag filament initiating at the bottom ITO electrode in the left device via the reduction process of Ag⁺ cations. Meanwhile, a new $V_{Br}$ conducting channel is also rebuilt (also from the bottom to the top electrode) through a similar reduction process ($V_{Br}^+ + e^- \rightarrow V_{Br}$). The current measured in region (I) is hence increased rapidly from $\sim 1 \times 10^{-7}$ to $\sim 2 \times 10^{-5}$ mA with the bias voltage, with the maximum current in this region restricted by the left device as its resistance is much larger than that of the right (i.e., equivalent to an ON-state RRAM in this stage).

While further increasing the applied voltage to phase into the region (II), the LEM current is suddenly switched and significantly increased to $\sim 4 \times 10^{-3}$ mA, resulting from the complete formation of the Ag filament and the reconstruction of $V_{Br}$ conducting channel in the left device. Here, both devices on either side of the LEM functions as the ON-state RRAM and are connected in series, rendering the maximum current (or the lowest resistance) of the LEM. Further ramping up the positive voltage to the region (III) causes both the Ag filament and the $V_{Br}$

conducting channel in the right device to start vanishing due to their associated oxidation processes, leading to an obvious current drop back down to ~$2 \times 10^{-5}$ mA. It shall be addressed that another $V_{Br}$ conducting channel can be induced and rebuilt (from the top to the bottom electrode) in the right device during this stage. Finally, a well rectifying diode-like behavior with a turn-on voltage of ~5 V is observed when the positive voltage reaches the region (IV), in which $Cs^+$ cations could be dissociated from the bromide vacancy as previous calculations have shown that $Cs^+$ cation has a much higher activation energy than other halide anions[31,47,48]. The respective repelling of $Cs^+$ cations and $Br^-$ anions towards the top Ag electrode and the bottom ITO results in the formation of p-i-n homojunction in the right device (while the left device remains as an ON-state RRAM), responsible for the subsequent EL emission observed in the LEM. The LEM has no EL emissions until the applied voltage reaches ~5 V, which is equivalent to the sum of the turn-on voltage of the individual LEC (~4.0 V, Fig. 2b) and the RRAM (~1.0 V, Fig. 4b), and that is reasonable when considering both devices are connected in series. Optimization of the device fabrication such as better control of the $CsPbBr_3$ QDs thin-film thickness and the employment of a new device scheme could further reduce the LEM turn-on voltage. When the applied voltage is swept back from 8 to 0 V, a similar electrical hysteresis effect is again observed (in reference to the previous discussion of Fig. 2c), validating that in the region (IV), the LEM is realized with the ON-state RRAM on the left connecting in series to the emitting LEC on the right. The negative part of the I-V characteristics exhibits the same behavior as the positive voltage, which is easily understood by the symmetry of the LEM device structure. It is this symmetry that enables one to modulate the functionality of each device in the LEM to be either an RRAM or an LEC by simple switching the sweep polarity applied to the LEM.

**Parallel optical and electrical readings of two-color emitting $CsPbBr_3$ QD-based LEM**. Finally, we test the operation speed of our all-perovskite LEM by applying a pulse voltage bias (pulse width 100 μs with 50% duty cycle), and reading the voltage response of the RRAM ($V_{RRAM}$) and the light emission of the LEC ($P_{LEC}$) in the LEM. As indicated by the circuit diagram in the top panel of the Fig. 7a, when the applied pulse voltage ($V_{in}$) is 6 V (that is, the LEM operates in the region (IV) of the Fig. 6), the negatively biased left device (denoted as Device 1) of the LEM behaves as an LEC, while the positively biased device on the right (denoted as Device 2) acts as an RRAM. Clearly, under this forward bias operation, both the $V_{RRAM}$ (middle panel) and the $P_{LEC}$ (bottom panel) traces keep up well with the $V_{in}$, demonstrating that our LEM can operate as fast as 5 kHz, at which the data stored in the LEM can be electrically read by the Device 2 and optically transmitted through the parallel light emission from the Device 1. As pointed out earlier that the speed of individual $CsPbBr_3$ QD-based LEC and RRAM depends strongly on the ion generation and movement in the perovskite QD active layer, the operation speed of our all-perovskite LEM could potentially be enhanced by improving the intrinsic transport properties of the perovskites[49,50]. The fast operation of our LEM remains if the polarity of the pulse voltage is reversed (see Fig. 7b), except for the fact that the Device 1 now becomes an RRAM with a negative voltage response and Device 2 functions as an LEC with a similar light emission power. Note that these two opposite situations of the LEM realized by simply changing the applied voltage polarity could represent two distinct states in the logic operation, and hence could be further utilized to enable the desirable multilevel data storage. Also note that such dual light-emitting and resistive-switching operations of the LEM with fast response could potentially implement a novel scheme for data encryption, where the light emission from the LEC is transmitted and detected by an optical sensor, and then the detected signal is converted into the encryption key for the user to access the encoded information recorded in the RRAM.

In addition to providing a parallel, non-contact optical reading, the emission color of the LEC can be employed as a real-time indicator of the LEM status (i.e., either in the "write" or the "erase" state). To this end, we experimentally realize a two-color emitting LEM by reducing the size of the perovskite QDs in the Device 2 to only ~20 nm for blue-shifting its emission wavelength due to the quantum confinement effect (Supplementary Fig. 2), while keeping the dots in the Device 1 the same (as those in the Fig. 1c). Under the forward ac operation (+6 V with a 250 μs pulse width), the Device 2 in the LEM plays the role of an ON-state RRAM, whereas the Device 1 operates as an emitting LEC with the emission wavelength of $\lambda_1 = 532$ nm (top panel in Fig. 7c). The write state of the LEM can thus be visually recognized by the greenish appearance of the Device 1. Upon reversing the ac bias voltage, the LEM turns to its erase mode, in which the Device 2 originally as an RRAM is switched off and becomes an LEC exhibiting the aqua luster, resulting from the emission of the smaller dots at $\lambda_2 = 515$ nm (bottom panel in Fig. 7c). As a result, one can "perceive" in real time (at the speed of 1 kHz) the electrically encoded digital status of the LEM, by using an optical filter to block one emission color and transmit the other, or using two optical detectors to separately register them from the LEM. Note that in the present proof-of-concept work, these two schemes are not experimentally pursued due to the limitation of our experimental setup, and we instead distinguish two colors of the LEM by differentiating their $P_{LEC}$ levels in a single detector (Fig. 7d). It is worth noting that this real-time recognition of the digital status of the LEM by simply observing its emission color is enabled by the fast switching of the single perovskite device, as demonstrated earlier in Fig. 5. Although the switching speed between two perovskite devices in the LEM is still slower than that of the state-of-the-art RRAM for performing data processing in series[51,52], we expect that it could be further improved by optimizing our perovskite synthesis and the perovskite device structure (such as device dimension, geometry, and layer thickness). Apart from the optimization on the device level, by employing the LEM in a more complicated network structure such as multicast mesh network[53,54], we envision that the overall transmitted data rate of the LEM could be further enhanced. In any case, the present demonstration on the fast operation and active switching of the all-perovskite LEM successfully sets a new benchmark for the development of more advanced LEM technologies. On a broader scale, this work also provides a new paradigm for generating novel device concepts and functionalities by employing the synergistic combination of electronic and photonic degrees of freedom in a single material system.

## Discussion

In conclusion, a fast, all-perovskite-based LEM is demonstrated for the first time by employing the outstanding electrical and optical properties of the $CsPbBr_3$ QDs in a simple Ag/PMMA/$CsPbBr_3$ QDs/ITO structure. The electrically driven ionic motions in the perovskite QD layer enable this structure to function either as a fast, reliable electrical memory, or a fast, efficient light source, and more importantly, the swift switching between the dual functionalities of the same structure by simply modulating its bias polarity. The highly-efficient optical transitions in the perovskite QD layer accompanied by the tunable emission colors not only allow for the realization of the LEM in the same material system through monolithic integration, but also the enhanced functionalities of the LEM to broaden the scope of its applications. We believe that this work could lead to more

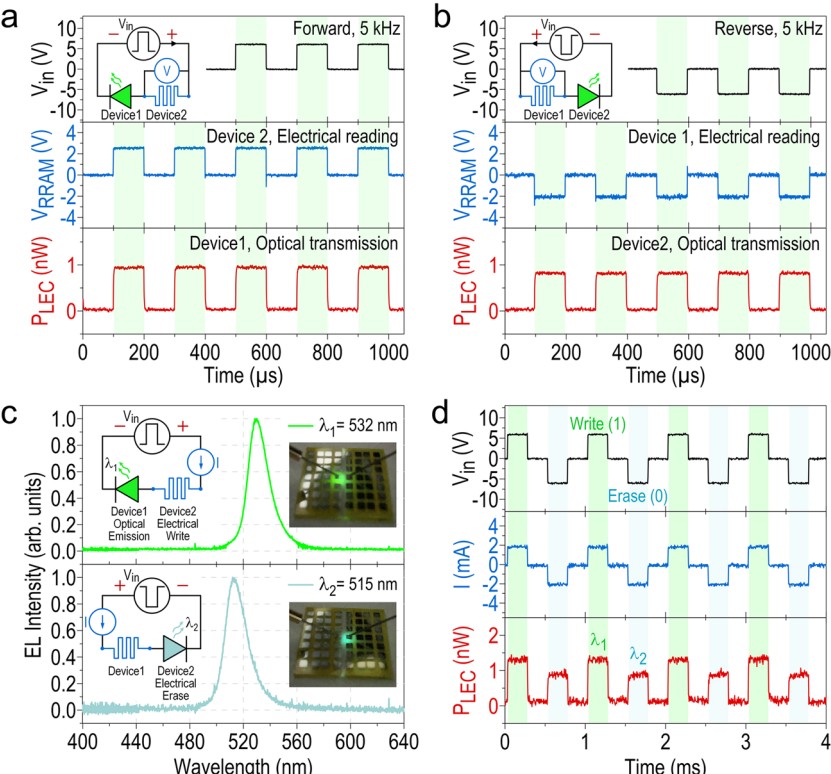

**Fig. 7 Parallel optical and electrical readings and two-color emissions of CsPbBr₃ QD-based LEM.** Time traces of $V_{RRAM}$ and $P_{LEC}$ of the all-perovskite LEM under **a** forward (+6 V) and **b** reverse (−6 V) pulse bias $V_{in}$ (100 µs width with 50% duty cycle) at 5 kHz, demonstrating the fast, parallel optical and electrical readings of the LEM regardless of the bias polarity. The equivalent circuit diagrams indicating the functionality of each device in the LEM under both bias polarities are also inserted in the top panels. **c** EL spectra of the two-color emitting all-perovskite LEM employing CsPbBr₃ QDs with two different sizes, one emitting at $\lambda_1$ = 532 nm (exhibiting green color, see the photograph inset in the top panel) while the other emitting at $\lambda_2$ = 515 nm (appearing aqua luster, see the photograph inset in the bottom panel). The corresponding circuit diagrams of the LEM when it appears green (top panel) or aqua (bottom panel) are also inserted. These two colors could serve as a real-time indicator of the digital status (i.e., write or erase) of the LEM. **d** Time traces of the current (I) and the emitting optical power ($P_{LEC}$) of the two-color emitting LEM when actively switching the $V_{in}$ polarity at 1 kHz. The difference in the $P_{LEC}$ levels of the two emission colors (bottom panel) are employed in this work to distinguish in real time the write and the erase state of the LEM.

powerful perovskite-based devices based on a more complicated device architecture, featuring high integration density, fast modulation speed, and multiple device functionality.

## Methods

**Synthesis of CsPbBr₃ QDs.** Lead (II) bromide (PbBr₂, 0.1845 g) and cesium bromide (CsBr, 0.1415 g) powders were dissolved in dimethyl sulfoxide (DMSO, 5 mL) to form the precursor solution. Oleic acid (OA, 0.5 mL), oleylamine (OAm, 0.25 mL), and hydrogen bromide (HBr, 5 µL) were then added to stabilize the precursor solution and kept at a stirring rate of 1000 rpm for 1 h. The mixed precursor solution was then poured into toluene (240 mL) to trigger the super-saturated recrystallization process for synthesizing the CsPbBr₃ QDs. After that, the entire solution was centrifuged at 1000 xg for 5 secs and we took its supernatant part by using a pipette to filter out the large particles. All of the above were conducted at room temperature.

**Deposition of Ag and ITO electrodes**. The top Ag (100 nm) and the bottom ITO (250 nm) electrodes were grown by using a RF sputtering system with the background chamber pressure of ~ $1.0 \times 10^{-6}$ torr. The working pressure and sputtering power were kept at ~$8.0 \times 10^{-3}$ torr and 50 W, respectively. The inert gas of argon at a flow rate of 6.0 and 10.0 sccm was introduced into the sputtering chamber while depositing Ag and ITO films, respectively. The deposition rate was ~ 4.0 Å/s for both the Ag and the ITO films.

**Measurement details**. EL spectra and intensities of the fabricated samples were measured by using a portable spectrometer (Ocean Optics USB4000), while its electrical characterizations were performed with a Keithley 2400 source meter. XRD patterns were measured by a multipurpose X-ray diffraction system (Bruker new D8 discover). Absorption spectra of the fabricated samples were obtained by the spectrophotometer (Aglient, Cary5000), and the PL spectra were measured by a portable spectrometer (Ocean Optics USB4000) with a 405 nm diode laser

excitation. Transient measurements of the LEC were conducted using a function generator (Agilent 33220 A), an oscilloscope (Keysight DSOX1204G), and a benchtop optical power meter (Newport 1936-R) with a photodiode (Newport 818-UV/DB) covering the spectral range of 200–1100 nm. Resistive switching of the RRAM was characterized using the same function generator and the oscilloscope with a 50 Ω load resistor for generating pulsed voltages and reading transient responses, respectively. Surface and cross-sectional morphologies of the fabricated samples were examined using field-emission SEM (JEOL, JSM7600F, 10 kV). TEM images and EDS spectra of the as-synthesized CsPbBr₃ QDs were performed by using an atomic-resolution electron microscope (JEOL, ARM200F, 200 kV).

## Data availability

The data that support the findings of this study are available from the corresponding authors on reasonable request.

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

## Acknowledgements

C.-C. C. and Y.-J. L. acknowledge the financial support from the Ministry of Science and Technology in Taiwan (contract Nos. MOST 108-2112-M-003-015-MY3 and MOST 109-2112-M-003-014-MY3). J.L. and K.T acknowledge the support of JSPS KAKENHI (Grant No. 19H05627). The authors would like to thank Mses. Su-Jen Ji and Chia-Ying Chien, and Prof. Chun-Hsien Chen of Ministry of Science and Technology (National Taiwan University) for technical support in SEM experiments and fruitful discussions.

## Author contributions

C.C.C., K.T., and Y.J.L. conceived and initiated the work. M.C.Y., C.J.L., T. H. C., C.C.C., and Y.J.L. designed the experiments. M.C.Y., C.J.L., K.H.L., and Y.P. conducted the device fabrication. M.C.Y., C.J.L., J.L., and T.H.C. assisted with the experiments and performed the measurements. M.C.Y., C.J.L., C.C.C., K.T., and Y.J.L. performed the data analysis. C.C.C., K.T., and Y.J.L. drafted the manuscript and supervised this work. All authors discussed the results and contributed to the manuscript.

## Competing interests

The authors declare no competing interests.
