## [Peer Review File · Nature Communications]

REVIEWER COMMENTS

Reviewer #1 (Remarks to the Author):

The authors demonstrated light emitting memory (LEM) devices based on inorganic CsPbBr₃ perovskite quantum dots (QDs). When two nominally identical Ag/CsPbBr₃/ITO devices are series-connected, one device can operate as a resistive random-access memory (RRAM) while the other simultaneously as a light-emitting electrochemical cell, or vice versa. The concept proposed by the authors are new and interesting, but overall contents are not enough to guarantee publication in the journal. Detailed comments are listed below.

1. There are reports on CsPbBr₃ QDs based resistive switching (RS) devices and light emitting diodes (LEDs) already. The performance of the RS device and LED in the work is relatively poor. I do not find novel approach to have improved performance for each device.
2. The RS switching speed has not been characterized with voltage pulses. Endurance of the RS devices should be provided. From the last figure, it can be estimated that the switching speed is a few seconds or hundreds of ms at least. Such slow speed is not appropriate for electronic and optical reading of the encoded information in communication and computation applications the authors suggested.
3. As the silver has been used for the top electrode, ion migration should be significant in the LED device, which can be seen in the Fig 2b and 2d. This affects retention and endurance performance of the device.
4. It is not clear whether the RS device has an initial forming process.
5. The top electrode is not transparent and the area of it must be small as possible for RS. Then, light emission to the top side is limited.

Again, the device performance has not been fully evaluated and it is not of high quality in terms of speed and endurance for possible applications the authors suggested.

Reviewer #2 (Remarks to the Author):

In this manuscript, authors demonstrated all-inorganic perovskite quantum dot light-emitting memories by manipulating the ion migrations in two identical and series-connected devices and exhibited that this system could realize simultaneous electronic and optical reading of the encoded information. They also utilized the movements of various ionic species and their reduction or oxidation processes to interpret the observed electronic and optical characteristics. The manuscript was generally well organized and well written. However, I do not think that it is suitable for publication in such high-quality journal but more suitable for other materials science journals. Below are some comments for authors' attention.

- (1) There are already some works about LEMs, how are they different from yours in terms of structure and performance or what are the distinct advantages of these new devices? Innovation of this work needs to be restated.
- (2) Authors claimed that the heating process would introduce undesirable thermal stress to the underlying ITO and reduce the electrical conductivity of ITO and degrade the overall LEM device performance. Experimental data about LEM devices utilizing bulk perovskites under the heating process should be provided for contrast. And how to probe the inherent quantum confinement effect on the device's performance?
- (3) For the RRAM device, does the oxygen vacancy induced by ITO contribute to the conduction mechanism? And what is the endurance capability, cycle to cycle and device to device variations of the device?
- (4) Can the proposed mechanism in Figure 4b be characterized and experimentally verified? And how to further decrease the processing time (2 s in Figure 5a) of this system?
- (5) The investigation about this LEM system is still relatively simple. Can authors propose several specific future potential applications of this system?

Reviewer #3 (Remarks to the Author):

This communication reports that all-inorganic CsPbBr₃ perovskite quantum dots (QDs) based Ag/CsPbBr₃/ITO device exhibited the characteristics of light-emitting as well as electrical memory characteristics using electrical field-induced ionic motions. The fabricated devices could act as all-perovskite light emitting memory device for simultaneous electronic and optical reading of the encoded information in communication and computation applications. The communication demonstrates a novel device concept, which could drive the new application of high performance perovskite quantum dots based optoelectronic devices. Here I have the following comments for the authors to address before further considering the acceptance of the manuscript:

1. In page 4, the authors described "The transfer of Cs⁺, Pb²⁺, and Br⁻ ions from the soluble to insoluble solvents is too fast during the supersaturated recrystallization process, which significantly enlarges the QD size to ~ 50 –100 nm, larger than the CsPbBr₃ QDs synthesized by the hot-injection method that generally have a smaller size of ~ 10 –15 nm". The size of perovskite QD significantly affected the electronic and optoelectronic characteristics and the long-term stability of the fabricated devices. Here I suggest the authors to discuss the device performance using different sizes of the QD.
2. The authors used poly(methyl methacrylate) (PMMA) as the protection layer on top of the CsPbBr₃ QDs. Why does it need the protection layer? Also, what is the criteria on the selection of the materials for the protection layer?
3. In figure 3d, the retention characteristic decreased from 10³ to 10² after 5000 seconds of operation. This is far below the standard compared to the reported resistive memory devices. I suggest the authors to improve the retention characteristics for the potential practical applications.
4. The authors showed green color light-emission in the fabricated devices. I suggest the authors to show the multi-color emission and demonstrate that the methodology could be applied to other kind of perovskite QD based devices.

RESPONSE TO REVIEWERS' COMMENTS

We would like to first thank the Editor and the Reviewers for careful reading of our manuscript and their valuable comments and suggestions. Particularly, we thank the positive comments from all three Reviewers. The first Reviewer stated that “*The concept proposed by the authors are new and interesting...*” The second Reviewer remarked that “*The manuscript was generally well organized and well written.*” The third Reviewer stated that “*The communication demonstrates a novel device concept, which could drive the new application of high performance perovskite quantum dots based optoelectronic devices.*” We are truly grateful and encouraged by these positive remarks. On the other hand, following the Reviewers’ other suggestions, we have significantly improved both the *content* and the *quality* of our work. Here we would like to highlight some of these improvements up front before carefully responding to the Reviewers’ questions and concerns point by point. These improvements are listed as follows in the order of their importance:

- (1) *The fast operation of the all-perovskite LEM at 5 kHz under both positive and negative pulsed biases:* we experimentally demonstrate that our all-perovskite LEM device can operate at a speed of as fast as 5 kHz, regardless of the polarity of the pulsed bias. This new result ensures the fast, parallel electrical and optical readings of the LEM desirable for practical applications. More details can be referred to Figs. 7a and b, as well as the corresponding text in the revised manuscript.
- (2) *The active switching between the RRAM and the LEC in an all-perovskite LEM device at 1 kHz:* this updated switching speed is much more enhanced as compared to that reported in the previous manuscript (switching time on the order of second). More importantly, this active switching greatly enhances the functionalities of the LEM device. More details can be referred to the data plotted in Fig. 7c and d, and the related text in the revised manuscript.
- (3) *The two-color emitting LEM enabling a real-time reading of its digital status (i.e., write or erase) by the emission colors:* we employ CsPbBr₃ QDs with two different sizes in a single LEM device, one corresponding to green emission and the other showing the aqua luster. This demonstration significantly broadens the scope of applications of the proposed all-perovskite LEM. More details on this new result can be referred to Figs. 7c and d, and the corresponding text in the revised manuscript.
- (4) *The fast switching of a single Ag/CsPbBr₃ QDs/ITO device between the RRAM and the LEC operations at 1 kHz by simply modulating its bias polarity:* this new data represents the first experimental demonstration of actively switching a single CsPbBr₃ QD-based device between its two fundamentally different functionalities

(i.e., light emission and data storage). More details can be found in Fig. 5 and the related text in the revised manuscript.

These new results that we are now presenting in the revised manuscript are *unprecedented*, to the best of our knowledge, and will provide a new paradigm for the development of more advanced LEM technologies based on a single material platform. Apart from these key results, in the revised manuscript we have also included several important new data, including:

(5) *The fast modulation of the optical emission from the perovskite LEC up to 80 kHz:* the fast speed in conjunction with the high efficiency of the *EL* emission guarantees that our perovskite LEC meets the stringent requirements of the LEM for potential applications in data communication and signal modulation. See Fig. 3 in the revised manuscript for more details.

(6) *The fast resistive-switching ($\sim 3.0 \mu\text{s}$) and high device stability of the perovskite RRAM with small device-to-device variations:* our perovskite RRAM can be swiftly switched between its HRS and LRS by a $\sim 3.0 \mu\text{s}$ -wide pulsed bias, and shows good retention and endurance performance. Our new study on the device-to-device variation of our RRAM also shows small deviations in both its set and reset voltages. See Fig. 4 and Supplementary Figures. 7 and 8 for more details.

It is very clear that, as compared to the data presented in the previous manuscript, the device performance of the perovskite LEC, the perovskite RRAM, and hence the all-perovskite based LEM is tremendously improved. *That is, not only is the device concept of the all-perovskite based LEM proposed in this work novel, but it can indeed be realized (again for the first time) with high device performance suitable for real-world applications.* Based on all these new and improved data with high quality and completeness, we have significantly revised our manuscript fully addressing the concerns and suggestions raised from all three Reviewers. With the novelty and the strong technical content presented therein, we truly believe that our revised manuscript meets the high standard for publication in Nature Communications.

In the following we address the reviewers' concerns point-by-point.

■ [Response to the comments of Reviewer 1]

The authors demonstrated light emitting memory (LEM) devices based on inorganic CsPbBr₃ perovskite quantum dots (QDs). When two nominally identical Ag/CsPbBr₃/ITO devices are series-connected, one device can operate as a resistive random-access memory (RRAM) while the other simultaneously as a light-emitting electrochemical cell, or vice versa. The concept proposed by the authors are new and interesting, but overall contents are not enough to guarantee publication in the journal. Detailed comments are listed below.

1. There are reports on CsPbBr₃ QDs based resistive switching (RS) devices and light emitting diodes (LEDs) already. The performance of the RS device and LED in the work is relatively poor. I do not find novel approach to have improved performance for each device.

Response: We appreciate this comment. However, the Reviewer may have missed the main point and more importantly, the novelty of our work. The present work is not meant to provide a new approach to realizing a *single* CsPbBr₃ QD-based RRAM and/or a *single* CsPbBr₃ QD-based LEC, and compare their performance separately against the current state-of-the art. Instead, *our work is focusing on the creation of a completely new device, that is, all-perovskite based LEM, by simply pairing two identical CsPbBr₃ QD-based devices in series.* We not only propose the concept of such a new device, but also implement the idea experimentally. To the best of our knowledge, our work is for the first time the demonstration of both the seamless integration and the active switching of two fundamentally different functionalities (i.e., light emission and data storage) in a single material (perovskite) system. As pointed out in the above highlights, the resulting all-perovskite based LEM can operate as fast as 5 kHz under both positive and negative pulsed biases, and shows a 1 kHz modulation speed when actively switching its RRAM and LEC by modulating the bias polarity. By employing QDs with two different sizes, we further demonstrate a two-color emitting LEM enabling the real-time reading of its digital status (i.e., write or erase) by the emission colors. These results make our all-perovskite based LEM highly promising for numerous practical applications, for example, in the data communication and signal processing, where its RRAM electrically reads the encoded information, and its LEC in parallel optically transmits the data to the distant. *We would like to emphasize that it is all these innovations that distinguish our work from all others.* We understand that the performance of the individual RRAM and LEC would affect that of an LEM one way or another, so we have significantly improved the performance of both devices, including their operation speed and reliability. The Reviewer can refer to Fig. 3 and 4, as well as Supplementary Figures 7 and 8 in the revised manuscript for more details.

2. The RS switching speed has not been characterized with voltage pulses. Endurance of the RS devices should be provided. From the last figure, it can be estimated that the switching speed is a few seconds or hundreds of ms at least. Such slow speed is not appropriate for electronic and optical reading of the encoded information in communication and computation applications the authors suggested.

Response: We have characterized the resistive switching speed of the RRAM with pulsed bias voltages by using a function generator (Agilent 33220A) and an oscilloscope (Keysight DSOX1204G) with a $50\ \Omega$ load resistor, for generating pulsed voltages and reading transient responses, respectively. The results are shown in Figs. 4d-f in the revised manuscript. It can be seen that that our perovskite RRAM can be swiftly switched between its HRS and LRS by a $\sim 3.0\ \mu\text{s}$ -wide pulsed bias, corresponding to a switching speed much faster than that estimated by the Reviewer (i.e., a few seconds or hundreds of ms at least). It is this fast switching speed that enables the fast operation of our LEM (speed = 5 kHz, see Fig. 7a and b in the main text), desirable for electronic and optical readings of the encoded information in the communication and computation applications. We also characterized its reliability performance, including retention, endurance, and device-to-device variation, and the corresponding results can be found in the revised manuscript and supplementary information (Supplementary Figures 7 and 8), all demonstrating the excellent reliability of our RRAM device. Accordingly, we have included new figures and made several edits in the manuscript:

(a) For demonstrating the switching speed of our RRAM under pulsed bias voltages, Figs. 4d-f and the following sentences (now in page 8, from line 17 to line 25) have been added in the revised manuscript.

Figure 4 d Circuit diagram of the measurement system for examining the resistive switching speed of the RRAM. e Transient responses of the RRAM during the set ($1.5\ \text{V}$, $3.8\ \mu\text{s}$) and reset ($-2.0\ \text{V}$, $3.8\ \mu\text{s}$) processes. Insets: enlarged time trace of the read pulses ($0.1\ \text{V}$, $3.8\ \mu\text{s}$) fed into the RRAM after set (lower left) and reset (upper right) pulse biases. f I_{RRAM} vs. pulse width of the applied bias V_{in} for both set (top) and reset (bottom) processes.

For practical applications, a fast switching speed of an RRAM is highly desirable. We characterize the switching speed of our CsPbBr₃ QD-based RRAM using a measurement setup (see Fig. 4d) composed of a pulse generator, an oscilloscope, and a load resistor (50 Ω). The transient switching between the HRS and the LRS, and the corresponding real-time current (IRRAM) are directly monitored by examining the bias change over the load resistor. As shown in Fig. 4e, read pulses (0.1 V, 3.8 μ s) are fed into the RRAM before and after the set/reset pulses. The RRAM is switched from the HRS to the LRS when a set pulse (1.5 V, 3.8 μ s) is applied, evidenced by the surge of IRRAM in response to the read pulse (inset, lower left). The RRAM is then switched back to the HRS upon applying a reset pulse (-2.0 V, 3.8 μ s), and therefore, almost no current is detected by the read pulse (inset, upper right). This fast switching behavior of our RRAM holds even when the pulse widths of the set (1.5 V)/reset (-2.0 V) bias are narrowed down to ~ 3.0 μ s, as can be seen in Fig. 4f.

(b) For proving the reliability performance of our RRAM, the following sentences (now in page 8, starting at line 25 all the way to the top of page 9) along with Supplementary Figures 7 and 8 and their descriptions have been added in the revised manuscript.

As far as the device reliability is concerned, both the HRS and the LRS (read at 0.1 V) of the RRAM can retain for over 10^5 sec and the ON/OFF current ratio keeps at $\sim 10^2$ (Supplementary Fig. 7a), indicating good retention stability. We also conduct an endurance test on our RRAM under a pulse sweeping mode (Supplementary Fig. 7b), and except for some fluctuations, the HRS/LRS ratio remains basically the same even after more than 10^4 sweeping cycles. Moreover, the RRAM can be switched reversibly between the HRS and the LRS more than 50 times when subject to repeated sweeping processes between a positive and a negative dc bias (Supplementary Fig. 7c), and its I - V characteristic shows a small device-to-device variation (Supplementary Fig. 8). All of these validate the reliable and reproducible write/erase characteristics of our CsPbBr₃ QD-based RRAM.

Supplementary Figure 7 | Retention and endurance tests of the RRAM mode of the CsPbBr₃ QD-based LEM

Supplementary Figure 7 a Retention performance of LRS and HRS of an Ag/CsPbBr₃ QDs/ITO device when acting as an RRAM. **b** Endurance test result (read at 0.1 V) of the device over 10⁴ cycles. **c** *I-V* characteristics of the device after different numbers of times of a dc bias sweep (0 V → +2 V → 0 V → -2 V → 0 V) up to 50 times.

Supplementary Figure 7a shows the retention performance of the RRAM mode of a single Ag/CsPbBr₃ QDs/ITO device by consecutively applying square-wave bias pulses (0.1 V on a 1.1 sec period) to read the RRAM. Both the HRS and the LRS can retain for over 10⁵ sec and the ON/OFF current ratio keeps at ~ 10², indicating good retention stability. Supplementary Figure 7b shows the endurance test results of the device under a modulated bias with a repetitive sequence of 1.5 V(Set)/ 0.1 V(Read)/ - 2.0 V(Reset)/ 0.1 V(Read). Except for some fluctuations in both HRS and LRS, there is no obvious decline of HRS/LRS ratio even after more than 10⁴ sweeping cycles. Moreover, as shown in Supplementary Figure 7c, the RRAM can be switched reversibly between its HRS and LRS more than 50 times by repeatedly sweeping a dc bias in a sequence of 0 V → +2 V → 0 V → -2 V → 0 V. The current ratio between HRS and LRS could maintain at ~ 10² during these alternating positive/negative dc bias sweeps. These results altogether ensure the reliable and reproducible write/erase characteristics of either one of the Ag/CsPbBr₃ QDs/ITO devices in our CsPbBr₃ QD-based LEM when acting as an RRAM.

Supplementary Figure 8 | Device-to-device variation in the set and reset voltages of the CsPbBr₃ QD-based RRAM

Supplementary Figure 8 a *I-V* characteristics of 18 CsPbBr₃ QD-based RRAM devices by setting the compliance current to $I_{CC} = 1$ mA. **b** Cumulative probability distributions of the set (right) and reset (left) voltages.

Supplementary Figure 8a shows *I-V* characteristics over 18 devices of CsPbBr₃ QD-based RRAM to estimate the substantial device-to-device variability in the set and reset voltages. Supplementary Figure 8b plots the cumulative probability distribution of the

set and reset voltages extracted from Supplementary Figure 8a over 18 devices. Accordingly, it is clear that the set process of the RRAM occurs mostly at the mean value of $\mu = 0.66$ V with a small standard deviation of $\sigma = 0.04$. The reset process, however, shows enhanced variability from the mean value of $\mu = -1.35$ V by a slightly larger standard deviation of $\sigma = 0.105$. The above observations validate that our CsPbBr₃ QD-based RRAM exhibits an acceptable device-to-device variability.

(c) For demonstrating the speed of our LEM, Figs. 7a and b and the following paragraph (now in page 11 and 12, starting at line 8 from the bottom of page 11 onward) have been added in the revised manuscript. Two new references, Refs. 49 and 50, are also added when discussing the speed of the LEM in the revised manuscript.

Figure 7 a and b Time traces of V_{RRAM} and P_{LEC} of the all-perovskite LEM under both forward (+ 6 V, left panel) and reverse (− 6 V, right panel) pulse bias V_{in} (100 μs width with 50% duty cycle) at 5 kHz, demonstrating the fast, parallel optical and electrical readings of the LEM regardless of the bias polarity. The equivalent circuit diagrams indicating the functionality of each device in the LEM under both bias polarities are also inserted in the top panels.

Finally, we test the operation speed of our all-perovskite LEM by applying a pulse voltage bias (pulse width 100 μs with 50% duty cycle), and reading the voltage response of the RRAM (V_{RRAM}) and the light emission of the LEC (P_{LEC}) in the LEM. As indicated by the circuit diagram in the top panel of the Fig. 7a, when the applied pulse voltage (V_{in}) is 6 V (that is, the LEM operates in the region (IV) of the Fig. 6), the negatively biased left device (denoted as *Device 1*) of the LEM behaves as an LEC, while the positively biased device on the right (denoted as *Device 2*) acts as an RRAM. Clearly, under this forward bias operation, both the V_{RRAM} (middle panel) and the P_{LEC} (bottom panel) traces keep up well with the V_{in} , demonstrating that our LEM can operate as fast as 5 kHz, at which the data stored in the LEM can be *electrically* read by the *Device 2* and *optically* transmitted through the parallel light emission from the *Device*

1. As pointed out earlier that the speed of individual CsPbBr₃ QD-based LEC and RRAM depends strongly on the ion generation and movement in the perovskite QD active layer, the operation speed of our all-perovskite LEM could potentially be enhanced by improving the intrinsic transport properties of the perovskites^{49,50}. The fast operation of our LEM remains if the polarity of the pulse voltage is reversed (see Fig. 7b), except for the fact that the *Device 1* now becomes an RRAM with a negative voltage response and *Device 2* functions as an LEC with a similar light emission power. Note that these two opposite situations of the LEM realized by simply changing the applied voltage polarity could represent two distinct states in the logic operation, and hence could be further utilized to enable the desirable multilevel data storage. Also note that such dual light-emitting and resistive-switching operations of the LEM with fast response could potentially implement a novel scheme for data encryption, where the light emission from the LEC is transmitted and detected by an optical sensor integrated with the LEM, and then converted into the encryption key for the user to access the encoded information recorded in the RRAM.

49. Begum, R., Parida, M. R., Abdelhady, A. L., Murali, B., Alyami, N. M., Ahmed, G. H., Hedhili, M. N., Bakr, O. M. & Mohammed, O. F. Engineering interfacial charge transfer in CsPbBr₃ perovskite nanocrystals by heterovalent doping. *J. Am. Chem. Soc.*, **139**, 731–737 (2017).

50. Chen, Z., Brocks, G, Tao, S. & Bobbert, P. A. Unified theory for light-induced halide segregation in mixed halide perovskites. *Nature. Comm.* **12**, 2687 (2021).

3. As the silver has been used for the top electrode, ion migration should be significant in the LED device, which can be seen in the Fig 2b and 2d. This affects retention and endurance performance of the device.

Response: As already mentioned in our response to the Reviewer's question #2, we have demonstrated good retention and endurance performance of our perovskite RRAM. The retention measurement result shows that the HRS and the LRS (read at 0.1 V) of the RRAM can retain for over 10⁵ sec and the ON/OFF current ratio keeps at ~ 10² (Supplementary Figure 7a). The endurance test of the RRAM under a pulse sweeping mode (Supplementary Figure 7b) reveals that, except for some fluctuations, the HRS/LRS ratio remains basically the same even after more than 10⁴ sweeping cycles. In addition to retention and endurance, we also show that the RRAM can be switched reversibly between the HRS and the LRS more than 50 times when subject to repeated sweeping processes between a positive and a negative dc bias (Supplementary Figure 7c), and its *I-V* characteristic shows a small device-to-device variation (Supplementary Figure 8).

4. It is not clear whether the RS device has an initial forming process.

Response: An initial electroforming process is indeed needed for resistive switching. The corresponding data, which in fact was already shown in Supplementary Figure 2 in the previous manuscript, is now presented in Supplementary Figure 9 in the revised manuscript. The electroforming voltage required for switching the LEM from HRS to LRS is about -4.5 V, and a two-step hopping in the measured current is observed during the electroforming process, implying two different types of conducting channels induced in the left and right device of the LEM.

Supplementary Figure 9 and the related descriptions shown below have been added in the Supplementary Information of the revised manuscript.

Supplementary Figure 9 | Electroforming process of the CsPbBr₃ QD-based LEM

Supplementary Figure 9 a Typical I - V curve of the electroforming process of the CsPbBr₃ QD-based LEM. **b** Illustration of the resistive switching behavior during the electroforming process in the left and right device of the LEM.

To obtain the resistive switching effect, an electroforming process is needed to perform on the fabricated LEM device. Supplementary Figure 9a shows the typical I - V curve of the electroforming process of our CsPbBr₃ QDs-based LEM. The electroforming voltage required for the LEM switching from the HRS to the LRS status is ~ -4.5 V, and a two-step hopping in the current is observed during the electroforming process (1st switch on at ~ -0.75 V), which suggests two different types of conducting channels are induced separately in the left and right device of the LEM. Supplementary Figure 9b depicts the possible scenario of the resistive switching behavior in the CsPbBr₃ QDs-based LEM during its electroforming process. After the electroforming process, both the Ag filament and V_{Br} conducting channel are well established in the right device. However, only the V_{Br} conducting channel is induced on the left since a negative bias is applied on its top Ag electrode which hinders the migration of Ag^+ cations over the active layer to form the filament.

5. The top electrode is not transparent and the area of it must be small as possible for RS. Then, light emission to the top side is limited.

Response: The area of the top Ag electrode is not an issue when considering light extraction. When characterizing our LEC, we collect the emissions *from its backside*, by placing the photodiode underneath the glass substrate, so that the reflected light by the top Ag electrode within the device can also be collected. This characterization scheme is in fact similar to the method for characterizing most of the organic LEDs. To clarify this, the following sentences (now in page 5, starting at line 5 from the bottom of the page) have been added to the revised manuscript.

In addition, similar to the method for characterizing most of the organic LEDs, the light emissions from our LEC is measured by a photodiode placed underneath the glass substrate, to ensure that the reflected light by the top Ag electrode within the device can also be collected.

6. Again, the device performance has not been fully evaluated and it is not of high quality in terms of speed and endurance for possible applications the authors suggested.

Response: The device performance in terms of speed and endurance has been fully evaluated. First, we have characterized the speed of the perovskite RRAM, the perovskite LEC, and the all-perovskite LEM. We have shown for individual devices that the RRAM can be swiftly switched between its HRS and LRS by a ~ 3.0 μs -wide pulsed bias (Fig. 4), and the optical emission from the LEC can be modulated at a speed of 80 kHz (Fig. 3). By combining the fast resistive switching and the fast light modulation, the all-perovskite LEM can operate at a speed of 5 kHz under both positive and negative pulsed biases (Fig. 7a and b). We have further shown that the RRAM and the LEC in an all-perovskite LEM device can be actively switched at 1 kHz. Second, we have demonstrated from the endurance test (Supplementary Figures 7) that the HRS/LRS ratio of the RRAM remains basically the same even after more than 10^4 sweeping cycles device under a modulated bias with a repetitive sequence of 1.5 V(Set)/0.1 V(Read) – 2.0 V(Reset) /0.1 V(Read). Based on the demonstrated high-quality device performance, the all-perovskite LEM is highly promising for applications that we refer to in the manuscript.

■ [Response to the comments of Reviewer 2]

In this manuscript, authors demonstrated all-inorganic perovskite quantum dot light-emitting memories by manipulating the ion migrations in two identical and series-connected devices and exhibited that this system could realize simultaneous electronic and optical reading of the encoded information. They also utilized the movements of various ionic species and their reduction or oxidation processes to interpret the observed electronic and optical characteristics. The manuscript was generally well organized and well written. However, I do not think that it is suitable for publication in such high-quality journal but more suitable for other materials science journals. Below are some comments for authors' attention.

1. There are already some works about LEMs, how are they different from yours in terms of structure and performance or what are the distinct advantages of these new devices? Innovation of this work needs to be restated.

Response: Our all-perovskite LEM differs significantly from previous works in terms of structure and performance. *First of all*, our LEM is realized based on a *single* material platform, by pairing two identical perovskite QD-based devices in series, whereas the LEMs reported in other works are based on the *heterogeneous* material integration. For instance, in Refs. 6 and 7, the LEM devices are formed by stacking a SiO₂-based memory on top of a GaN-based light emitting diode. These vertically integrated structures for LEM are fairly complicated, and careful considerations must be made to ensure the proper device function. In these LEM structures, both the memory material and the common electrode that connects the memory to the underlying light emitter have to be transparent to the optical emission. The common electrode (graphene sheet in these cases) needs to have a high electrical conductivity and more importantly to provide good current spreading. An interfacial layer of metal nanoparticles is also needed to enhance the ON/OFF ratio of the memory. All of these requirements not only pose a serious constraint on the material selection and compatibility, but significantly add the complexity in the device fabrication. Our all-perovskite LEM, on the contrary, series connect the perovskite RRAM *laterally* to the perovskite LEC on the same glass substrate, which minimizes not only the absorption of the optical emission by the memory atop, but also the reflection due to the impedance mismatch between the two very dissimilar material systems in the vertical structure. The all-perovskite LEM also eliminates the need of transparent common electrodes. Moreover, the fabrication of the all-perovskite LEM relies primarily on a simple spin-casting process, which is much simpler than that employed in the aforementioned works. *Second*, our all-perovskite LEM promises much more device functionalities than all the other works (Refs. 5-8). We have demonstrated that, enabled by the integration of the two identical devices, a swift operation as fast as 5 kHz can be achieved under both positive and negative pulsed

biases applied on the all-perovskite LEM. The two opposite situations of the LEM realized by the active switching could represent two distinct states in the logic operation, and hence could be further utilized to enable the desirable multilevel data storage. This fast, active switching further enables the two-color emitting LEM that we have demonstrated (Fig. 7 c and d), by employing QDs with two different sizes in the LEM. This two-color emitting LEM allows for a real-time reading of its digital status (i.e., write or erase) readily by the emission colors. All of these new functionalities of our all-perovskite LEM demonstrated in this work are *unprecedented*, and cannot be achieved neither in the aforementioned vertically integrated LEM employing two dissimilar material systems (Refs. 6 and 7), nor the single silicon nitride-based (Ref. 8) or polymer bistable (Refs. 4 and 5) light-emitting device. *Finally*, it is worth noting that none of these early demonstrations shows the operation speed of their LEM devices except for Ref. 8 (switching time = 5.8 sec), which makes it unclear whether these devices can indeed be deployed for the real-world applications. Therefore, *it is these innovations and advantages that make our all-perovskite based LEM unique and superior to other works.*

For clearly stating the novelty of our LEM, in the revised manuscript we have added the following paragraph, now in page 2 and the top of page 3, starting at line 9 from the bottom of page 2, which reads,

For instance, in Refs. 6 and 7, the LEM devices are formed by stacking a SiO₂-based memory on top of a GaN-based light emitting diode. These vertically integrated structures for LEM are fairly complicated, and careful considerations must be made to ensure the proper device function. In these LEM structures, both the memory material and the common electrode that connects the memory to the underlying light emitter have to be transparent to the optical emission. The common electrode (graphene sheet in these cases) needs to have a high electrical conductivity and more importantly to provide good current spreading. An interfacial layer of metal nanoparticles is also needed to enhance the ON/OFF ratio of the memory. All of these requirements not only pose a serious constraint on the material selection and compatibility, but significantly add the complexity in the device fabrication. Also, none of these demonstrations shows the operation speed of their LEM devices, which makes it unclear whether these devices can indeed be deployed for the real-world applications.

We have also added a sentence, now in page 3, starting at line 9 from the bottom of the page, which reads,

All of these new functionalities of our all-perovskite LEM demonstrated in this work are unprecedented, and cannot be achieved neither in the aforementioned vertically

integrated LEM employing two dissimilar material systems^{6,7}, nor the single silicon nitride-based⁸ or polymer bistable^{4,5} light-emitting device.

Lastly, we have added the following sentences (now in page 3, starting at line 15 from the bottom of the page), which read,

By taking advantage of the fast, electrically switchable ionic motions and the corresponding bifunctionalities (that is, RRAM and LEC) in a single perovskite structure, here we propose and demonstrate a novel all-inorganic perovskite CsPbBr₃ LEM by monolithically integrating a perovskite LEC with a perovskite RRAM. We show that this all-perovskite LEM could electrically read the encoded data by its RRAM and optically transmit the information through the emission from its LEC at a speed of 5 kHz. We further demonstrate a two-color emitting LEM by employing QDs with two different sizes in the device, enabling the real-time reading of the LEM digital status (either write or erase) simply by its emission colors.

2. Authors claimed that the heating process would introduce undesirable thermal stress to the underlying ITO and reduce the electrical conductivity of ITO and degrade the overall LEM device performance. Experimental data about LEM devices utilizing bulk perovskites under the heating process should be provided for contrast. And how to probe the inherent quantum confinement effect on the device's performance?

Response: We have investigated the effect of heating during the device fabrication on the LEM device performance by comparing two CsPbBr₃ RRAM devices, one with a CsPbBr₃ QD active layer, the other with a bulk CsPbBr₃ active layer grown in a high-temperature (650 °C) furnace system. Both devices have basically the same active-layer thickness (~ 1 μm, see Supplementary Figure 1a and b). From their *I-V* characteristics (Supplementary Figure 1d), it is clear that both the set (1.52 V) and reset (− 2.64 V) voltages for the RRAM device using the CsPbBr₃ bulk active layer are higher than those for the QD-based device (set: 0.82 V, reset: − 1.12 V). The ON/OFF current ratio of the CsPbBr₃ bulk-based RRAM is also reduced to be ≤ 10. These observations agree well with the degradation of the electrical conductivity of the ITO electrodes after subject to the thermal treatment (Supplementary Figure 1c). Therefore, the CsPbBr₃ QDs are preferred over the bulk in this work as the active layer, since the entire process for synthesizing the CsPbBr₃ QDs is conducted at room temperature, and thus the possible influence of thermal stress on the RRAM performance, such as the increased resistivity of the underlying patterned ITO electrodes clearly shown here, can be eliminated.

On the other hand, to explore the tunable bandgaps of the CsPbBr₃ resulting from the quantum confinement effect and their importance in the CsPbBr₃ QD-based LEM functionalities, we systematically reduce the size of the CsPbBr₃ QDs by changing the

mixture ratios between the Oleic acid (*OA*) and the oleylamine (*OAm*) during the stabilization process for the $\text{PbBr}_2/\text{CsBr}$ precursor solution. The average sizes of the synthesized CsPbBr_3 QDs (all with the same stoichiometry) are 123, 63, and 20 nm, for the *OA*: *OAm* ratios of 0.5 ml: 0.25 ml, 1.0 ml: 1.0 ml, and 1.5 ml:1.5 ml, respectively (Supplementary Figure 2a). The size distributions of the synthesized CsPbBr_3 QDs (in percentage) are also estimated and inserted in the figures. Clearly, gradually reducing the QD size blueshifts the peak emission wavelength of the *PL* spectra, from 526 to 520, and then 511 nm, due to the increased quantum confinement effect (Supplementary Figure 2b and 2c). The demonstrated tunable bandgaps of the CsPbBr_3 QDs offer a great opportunity to significantly enhance the LEM device functionalities, as evidenced by the realization of the two-color emitting LEM device employing two different QDs (see Figure 7 in the main text).

For proving the adverse heating effect in the preparation of bulk CsPbBr_3 , we have included the relevant data (both cross-sectional SEMs and measured *I-V* characteristics) in Supplementary Figure 1 in the Supplementary Information of the revised manuscript.

Supplementary Figure 1 | CsPbBr_3 bulk-based RRAM device

Supplementary Figure 1 Cross-sectional SEM image of the RRAM device based on a **a** bulk and a **b** quantum-dot perovskite active layer, respectively. **c** Measured resistivity of the patterned ITO electrodes with (black line) and without (red line) the thermal treatment at 650 °C for 10 mins. **d** *I-V* characteristics of the RRAM with a bulk (black curve) and a quantum-dot (red curve) active layer.

Supplementary Figure 1a shows the cross-sectional SEM image of the CsPbBr_3 RRAM employing a bulk perovskite active layer grown by a vapor phase approach in a furnace system. The glass substrate (area = 1.5 cm × 1.5 cm) with patterned ITO electrodes is

placed downstream of a quartz tube located in the furnace center. Two alumina boats loaded respectively with PbBr₂ and CsBr powders as precursors are put in the heating center of the tube. Prior to heating, the quartz tube is pumped down to 0.02 Torr by a mechanical pump, and then purged with high-purity nitrogen at a flow rate of 50 sccm to maintain a pressure of 600 Torr. The furnace is then heated to 650 °C in ~ 30 mins and stayed there for 10 mins. After that, the furnace is cooled down naturally to room temperature before removing the sample. As shown in the SEM, the grown CsPbBr₃ bulk is ~ 1 μm thick. The PMMA layer and the Ag electrode are subsequently deposited on top of the CsPbBr₃ bulk layer to complete the RRAM fabrication. For comparison, a CsPbBr₃ QD-based RRAM is also prepared at room temperature (the QD layer thickness is also ~ 1 μm), with its cross-sectional SEM image shown in Supplementary Figure 1b. As the measured resistivity of the patterned ITO electrodes increases from $\rho = 2.07 \times 10^{-4}$ to $4.60 \times 10^{-3} \Omega \cdot cm$ after the thermal treatment (Supplementary Figure 1c), both the set (1.52 V) and reset (– 2.64 V) voltages for the RRAM device using the CsPbBr₃ bulk active layer are higher than those for the QD-based device (set: 0.82 V, reset: – 1.12 V, Supplementary Figure 1d). The ON/OFF current ratio of the CsPbBr₃ bulk-based RRAM is also reduced to be ≤ 10 . Therefore, the CsPbBr₃ QDs are preferred over the bulk in this work as the active layer, since the entire process for synthesizing the CsPbBr₃ QDs is conducted at room temperature, and thus the possible influence of thermal stress on the RRAM performance, such as the increased resistivity of the underlying patterned ITO electrodes clearly shown here, can be eliminated.

For demonstrating the tunable bandgaps of the CsPbBr₃ resulting from the quantum confinement effect and their importance in the CsPbBr₃ QD-based LEM functionalities, we have included the relevant data in Supplementary Figure 2 in the Supplementary Information of the revised manuscript.

Supplementary Figure 2 | Quantum confinement effect of the CsPbBr₃ QDs

Supplementary Figure 2 a Top-view SEM images of the CsPbBr₃ QDs with different sizes obtained by varying the OA: OAm ratios of the stabilization solution during the synthesis. The size distributions of the synthesized dots (in percentage) are also inserted in the images. The scale bars are 200 nm in all SEMs. **b** PL spectra of the synthesized CsPbBr₃ QDs with different sizes. **c** Size dependence of the CsPbBr₃ QDs on the peak emission wavelength of their PL spectra.

To explore the tunable bandgaps of the CsPbBr₃ resulting from the quantum confinement effect and their importance in the CsPbBr₃ QD-based LEM functionalities, we systematically reduce the size of the CsPbBr₃ QDs by changing the mixture ratios between the Oleic acid (OA) and the oleylamine (OAm) during the stabilization process for the PbBr₂/CsBr precursor solution. The average sizes of the synthesized CsPbBr₃ QDs (all with the same stoichiometry) are 123, 63, and 20 nm, for the OA: OAm ratios of 0.5 ml: 0.25 ml, 1.0 ml: 1.0 ml, and 1.5 ml:1.5 ml, respectively (Supplementary Figure 2a). The size distributions of the synthesized CsPbBr₃ QDs (in percentage) are also estimated and inserted in the figures. Clearly, gradually reducing the QD size blueshifts the peak emission wavelength of the PL spectra, from 526 to 520, and then 511 nm, due to the increased quantum confinement effect (Supplementary Figure 2b and 2c). The demonstrated tunable bandgaps of the CsPbBr₃ QDs offer a great opportunity to significantly enhance the LEM device functionalities, as evidenced by the realization of the two-color emitting LEM device employing two different QDs (see Figure 7 in the main text).

We have also made edits to a few sentences, and they are now in page 4, from line 9 to line 15, which read,

Here the CsPbBr₃ QDs are preferred than bulk perovskites for two reasons. First, in order to form the latter, either the perovskite precursor solution^{26–28} or the mixed

powers of PbBr₂ and CsBr²⁹⁻³¹ would need to be subject to a period of high temperature for facilitating the nucleation and crystallization of the bulk perovskites, but the heating process would introduce undesirable thermal stress to the underlying ITO and reduce the electrical conductivity of ITO^{32,33} and degrade the overall LEM device performance (Supplementary Fig. 1). Moreover, the perovskites QDs would offer a great opportunity compared to its bulk counterpart utilizing their inherent quantum confinement effect to significantly enhance the LEM device functionalities (Supplementary Fig. 2).

Finally, we have also added three new references, and they are now Refs. 29-31 in the revised manuscript.

29. Hu, X. et al. Direct vapor growth of perovskite CsPbBr₃ nanoplate electroluminescence devices. *ACS Nano* **11**, 9869-9876 (2017).
30. Wang, Y., Yang, F., Li, X., Ru, F., Liu, P., Wang, L., Ji, W., Xia, J., & Meng, X. Epitaxial growth of large-scale orthorhombic CsPbBr₃ perovskite thin films with anisotropic photoresponse property. *Adv. Func. Mater.* **29**, 1904913 (2019).
31. Wang, Y., Jia, C., Fan, Z., Lin, Z., Lee, S.-J., Atallah, T. L., Caram, J. R., Huang, Y., & Duan, X. Large-area synthesis and patterning of all-inorganic lead halide perovskite thin films and heterostructures. *Nano Lett.* **21**, 1454-1460 (2021).

3. For the RRAM device, does the oxygen vacancy induced by ITO contribute to the conduction mechanism? And what is the endurance capability, cycle to cycle and device to device variations of the device?

Response: There is no clear evidence that the oxygen vacancy induced by the ITO electrode contributes to the conduction mechanism in our CsPbBr₃ QD-based RRAM devices. We fabricate two CsPbBr₃ QD-based RRAM devices, one with ITO and the other with Pt as the bottom electrode. Their *I-V* characteristics (see Supplementary Figure 6), including the set/reset voltages and the current ratio between the HRS and the LRS are essentially the same. It can be inferred from these results that there are negligible oxygen ions induced within the CsPbBr₃ QD active layer under an external bias. This is in a sharp contrast to the RRAM using oxides such as SiO_x (*ECS J. Solid State Sci. Technol.* **5**, Q115 (2016)), HfO_x (*IEEE Trans Electron Devices* **66**, 1276 (2019)), or AlO_x (*Sci. Rep.* **6**, 29347 (2016)) as its active layer materials, where the oxygen ions (O²⁻) generated in the oxide active layer during the dissociation process are easily propelled into the ITO electrode under a bias, which subsequently results in both the built-up of the oxygen ion reservoir due to the intrinsic oxygen-vacancy-rich nature of the ITO, and the enhanced stability of the conduction properties of the RRAM. On the other hand, we have characterized the device reliability performance of our RRAM devices. As can be seen in Supplementary Figure 7, both the HRS and the LRS

(read at 0.1 V) of the RRAM can retain for over 10^5 sec and the ON/OFF current ratio keeps at $\sim 10^2$ (Supplementary Figure 7a), indicating good retention stability. We also conduct an endurance test on our RRAM under a pulse sweeping mode (Supplementary Figure 7b), and except for some fluctuations, the HRS/LRS ratio remains basically the same even after more than 10^4 sweeping cycles. Moreover, the RRAM can be switched reversibly between the HRS and the LRS more than 50 times when subject to repeated sweeping processes between a positive and a negative dc bias (Supplementary Figure 7c), and its I - V characteristic shows a small device-to-device variation (Supplementary Figure 8). All of these validate the reliable and reproducible write/erase characteristics of our CsPbBr₃ QD-based RRAM.

We have included the relevant data in Supplementary Figure 6 in the Supplementary Information of the revised manuscript.

Supplementary Figure 6 | Influence of the ITO electrode on the resistive switching characteristic of the CsPbBr₃ QD-based RRAM

Supplementary Figure 6 a I - V characteristics of the CsPbBr₃ QD-based RRAM with Pt (black curve) and ITO (red curve) as the bottom electrode, respectively (compliance current to $I_{CC} = 1$ mA). **b** I - V measurement data for determining the electrical resistivity of the ITO (black line) and the Pt (red line) electrodes.

Supplementary Figure 6a shows the I - V characteristics of the CsPbBr₃ QD-based RRAM with Pt (black curve) and ITO (red curve) as the bottom electrode when applying a positive poling voltage (that is, Ag is the anode with Pt being the cathode). Although the Pt electrode shows a lower electrical resistivity than that of the patterned ITO electrode (see Supplementary Figure 6b), both devices exhibit almost identical I - V characteristics with the same current ratio between the HRS and the LRS of $\sim 10^2$. The set and reset voltages for the Ag/PMMA/CsPbBr₃ QD/Pt device are ~ 0.78 and ~ -1.20 V, respectively, which are similar to those of the reference device with the ITO bottom electrode (set voltage ~ 0.82 V, reset voltage ~ -1.12 V), and also comparable

to the reported data of the perovskite RRAM using the Pt electrode [Adv. Electron. Mater. 1900754 (2019)]. Most importantly, the above observation confirms that the intrinsic oxygen-vacancy-rich nature inherited in the ITO electrodes barely affects the resistive switching characteristics of the CsPbBr₃ QD-based RRAM.

For proving the reliability performance of our RRAM, the following sentences (now in page 8, starting at line 25 all the way to the top of page 9) along with Supplementary Figures 7 and 8 and their descriptions have been added in the revised manuscript.

As far as the device reliability is concerned, both the HRS and the LRS (read at 0.1 V) of the RRAM can retain for over 10⁵ sec and the ON/OFF current ratio keeps at ~ 10² (Supplementary Fig. 7a), indicating good retention stability. We also conduct an endurance test on our RRAM under a pulse sweeping mode (Supplementary Fig. 7b), and except for some fluctuations, the HRS/LRS ratio remains basically the same even after more than 10⁴ sweeping cycles. Moreover, the RRAM can be switched reversibly between the HRS and the LRS more than 50 times when subject to repeated sweeping processes between a positive and a negative dc bias (Supplementary Fig. 7c), and its I-V characteristic shows a small device-to-device variation (Supplementary Fig. 8). All of these validate the reliable and reproducible write/erase characteristics of our CsPbBr₃ QD-based RRAM.

Supplementary Figure 7 | Retention and endurance tests of the RRAM mode of the CsPbBr₃ QD-based LEM

Supplementary Figure 7 a Retention performance of LRS and HRS of an Ag/CsPbBr₃ QDs/ITO device when acting as an RRAM. **b** Endurance test result (read at 0.1 V) of the device over 10⁴ cycles. **c** I-V characteristics of the device after different numbers of times of a dc bias sweep (0 V → +2 V → 0 V → -2 V → 0 V) up to 50 times.

Supplementary Figure 7a shows the retention performance of the RRAM mode of a single Ag/CsPbBr₃ QDs/ITO device by consecutively applying square-wave bias pulses (0.1 V on a 1.1 sec period) to read the RRAM. Both the HRS and the LRS can retain for over 10⁵ sec and the ON/OFF current ratio keeps at ~ 10², indicating good retention stability. Supplementary Figure 7b shows the endurance test results of the device under

a modulated bias with a repetitive sequence of 1.5 V(Set)/ 0.1 V(Read)/ – 2.0 V(Reset)/ 0.1 V(Read). Except for some fluctuations in both HRS and LRS, there is no obvious decline of HRS/LRS ratio even after more than 10^4 sweeping cycles. Moreover, as shown in Supplementary Figure 7c, the RRAM can be switched reversibly between its HRS and LRS more than 50 times by repeatedly sweeping a dc bias in a sequence of 0 V → +2 V → 0 V → –2 V → 0 V. The current ratio between HRS and LRS could maintain at $\sim 10^2$ during these alternating positive/negative dc bias sweeps. These results altogether ensure the reliable and reproducible write/erase characteristics of either one of the Ag/CsPbBr₃ QDs/ITO devices in our CsPbBr₃ QD-based LEM when acting as an RRAM.

Supplementary Figure 8 | Device-to-device variation in the set and reset voltages of the CsPbBr₃ QD-based RRAM

Supplementary Figure 8 a *I-V* characteristics of 18 CsPbBr₃ QD-based RRAM devices by setting the compliance current to $I_{CC} = 1$ mA. **b** Cumulative probability distributions of the set (right) and reset (left) voltages.

Supplementary Figure 8a shows the *I-V* characteristics of total 18 CsPbBr₃ QD-based RRAM devices to study the device-to-device variability in their set and reset voltages. Supplementary Figure 8b plots the cumulative probability distribution of the set and reset voltages extracted from Supplementary Figure 8a. It is clear that the set process of the RRAMs occurs mostly at the mean value of $\mu = 0.66$ V with a small standard deviation of $\sigma = 0.04$, though the reset process shows a slightly larger variability ($\sigma = 0.105$) from the mean value of $\mu = -1.35$ V. The above observations validate that our CsPbBr₃ QD-based RRAM exhibits a good device-to-device variability.

We have also added a new sentence, now in page 8, at the end of the first paragraph, which reads,

Note that the intrinsic oxygen-vacancy-rich nature inherited in the ITO electrode barely affects the resistive switching characteristics of the CsPbBr₃ QD-based RRAM, as illustrated in Supplementary Fig. 6.

4. Can the proposed mechanism in Figure 4b be characterized and experimentally verified? And how to further decrease the processing time (2 s in Figure 5a) of this system?

Response: Yes, it can. We investigate the ion migrations responsible for the observed *I-V* characteristics in the LEM by using the SEM measurement and EDS analysis. Supplementary Figure 5 shows the cross-sectional SEM images of the Ag/PMMA/CsPbBr₃ QDs/ITO stack **a** without and **b** with a positive poling bias of 2 V (Ag: anode, ITO: cathode). Cross-sectional distributions of Ag (red curve) and Br (green curve) are plotted in both figures to elucidate the migration tendency of the dissociated cations and anions inside the CsPbBr₃ QD layer. For both samples, four different locations denoted as 1 – 4 from the top to the bottom electrode are selected to qualitatively inspect the changes of the atomic ratios (including Ag: Pb and Br: Pb) due to the poling bias, and the results are also plotted as column graphs inserted in the figures. For the sample without the poling bias (Supplementary Figure 5a), the highest density of Ag is found at the location 1 near the top electrode (Ag: Pb ratio ~ 20.08), and the detected Ag signal decreases gradually along the detecting depth, corresponding to a monotonous decline of the Ag: Pb ratio from ~0.58 (location 2), ~0.21 (location 3) to ~0.03 (location 4). After applying the positive poling bias on the RRAM (Supplementary Figure 5b), the detected Ag signal basically follows a similar trend, except that in the location 4 close to the device bottom, a high Ag density with an enhanced Ag: Pb ratio of ~1.20 is detected. It validates that the Ag⁺ does migrate over the CsPbBr₃ QD active layer and then oxidize to form Ag clusters (Ag⁺ + e⁻ → Ag) at the bottom ITO cathode.

As for the migration tendency of Br, both samples exhibit fairly similar distributions from the location 2 onwards, showing a stable Br: Pb ratio of ~3.30. However, a significant difference between two samples is observed in the location 1. This disparity can be easily understood, as for an RRAM using Ag as the anode, the top Ag electrode has a high electrochemical activity and can preserve the Br⁻ ions to form AgBr_x (Ag + Br⁻ → AgBr + e⁻), so that a much higher amount of Br (the Br: Pb ratio ~ 6.91) is expected in the location 1. In parallel, the bromide vacancy (V_{Br}⁺) is formed inside the CsPbBr₃ QD active layer after Br⁻ anions are impelled toward the top Ag electrode under the bias. V_{Br}⁺ also induces the construction of the conducting channel (V_{Br}⁺ + e⁻ → V_{Br}) and hence the subsequent transition from the HRS to the LRS in the *I-V* characteristics.

As a result, under the positive poling bias both the V_{Br} conducting channel and the Ag filament are well established in the RRAM and responsible for its resistive switching. Thus, the observation that much higher amounts of Ag and Br are found respectively in the location 4 and 1, is the direct evidence to prove the generation of the cations and anions, and how they are driven inside the $CsPbBr_3$ QDs by the external field. More importantly, based on the result here, the mechanism for the dynamic ionic transport and conduction processes in the perovskite LEM can be clearly outlined, as elucidated in Fig. 6b in the main text.

Supplementary Figure 5 and the related descriptions shown below have been added in the Supplementary Information of the revised manuscript.

Supplementary Figure 5 | Migration tendency of Ag and Br elements in the $CsPbBr_3$ QD-based RRAM

Supplementary Figure 5 Migration tendency of Ag and Br observed in the cross-sectional SEM images of the Ag/PMMA/ $CsPbBr_3$ QDs/ITO stack **a** without and **b** with a positive poling bias of 2 V. Cross-sectional distributions of Ag (red curve) and Br (green curve) from top to bottom are also plotted in both figures. Insets: column graphs of the Ag: Pb and Br: Pb ratios measured at different locations denoted as 1, 2, 3, and 4.

We investigate the ion migrations responsible for the observed I - V characteristics in the LEM by using the SEM measurement and EDS analysis. Supplementary Figure 5 shows the cross-sectional SEM images of the Ag/PMMA/ $CsPbBr_3$ QDs/ITO stack **a** without and **b** with a positive poling bias of 2 V (Ag: anode, ITO: cathode). Cross-sectional distributions of Ag (red curve) and Br (green curve) are plotted in both figures to elucidate the migration tendency of the dissociated cations and anions inside the $CsPbBr_3$ QD layer. For both samples, four different locations denoted as 1 – 4 from the top to the bottom electrode are selected to qualitatively inspect the changes of the atomic ratios (including Ag: Pb and Br: Pb) due to the poling bias, and the results are also plotted as column graphs inserted in the figures. For the sample without the poling

bias (Supplementary Figure 5a), the highest density of Ag is found at the location 1 near the top electrode (Ag: Pb ratio ~ 20.08), and the detected Ag signal decreases gradually along the detecting depth, corresponding to a monotonous decline of the Ag: Pb ratio from ~0.58 (location 2), ~0.21 (location 3) to ~0.03 (location 4). After applying the positive poling bias on the RRAM (Supplementary Figure 5b), the detected Ag signal basically follows a similar trend, except that in the location 4 close to the device bottom, a high Ag density with an enhanced Ag: Pb ratio of ~1.20 is detected. It validates that the Ag^+ does migrate over the CsPbBr_3 QD active layer and then oxidize to form Ag clusters ($\text{Ag}^+ + \text{e}^- \rightarrow \text{Ag}$) at the bottom ITO cathode.

As for the migration tendency of Br, both samples exhibit fairly similar distributions from the location 2 onwards, showing a stable Br: Pb ratio of ~3.30. However, a significant difference between two samples is observed in the location 1. This disparity can be easily understood, as for an RRAM using Ag as the anode, the top Ag electrode has a high electrochemical activity and can preserve the Br^- ions to form AgBr_x ($\text{Ag} + \text{Br}^- \rightarrow \text{AgBr} + \text{e}^-$), so that a much higher amount of Br (the Br : Pb ratio ~ 6.91) is expected in the location 1. In parallel, the bromide vacancy (V_{Br}^+) is formed inside the CsPbBr_3 QD active layer after Br^- anions are impelled toward the top Ag electrode under the bias. V_{Br}^+ also induces the construction of the conducting channel ($\text{V}_{\text{Br}}^+ + \text{e}^- \rightarrow \text{V}_{\text{Br}}$) and hence the subsequent transition from the HRS to the LRS in the I - V characteristics.

As a result, under the positive poling bias both the V_{Br} conducting channel and the Ag filament are well established in the RRAM and responsible for its resistive switching. Thus, the observation that much higher amounts of Ag and Br are found respectively in the location 4 and 1, is the direct evidence to prove the generation of the cations and anions, and how they are driven inside the CsPbBr_3 QDs by the external field. More importantly, based on the result here, the mechanism for the dynamic ionic transport and conduction processes in the perovskite LEM can be clearly outlined, as elucidated in Fig. 6b in the main text.

On the other hand, we have significantly improved our perovskite-based RRAM and LEC devices for further decreasing the processing time of the all-perovskite LEM for dual optical and electrical readings. We have demonstrated that our perovskite RRAM can be swiftly switched between its HRS and LRS by a ~ 3.0 μs -wide pulsed bias (Fig. 4), and the optical emission from the perovskite LEC can be modulated as fast as 80 kHz (Fig. 3). By combining the fast resistive switching and the fast light modulation, the all-perovskite LEM can operate at a speed of 5 kHz under both positive and negative pulsed biases (Fig. 7a and b). We have further shown that the RRAM and the LEC in

an all-perovskite LEM device can be actively switched at 1 kHz, which is much faster than the speed (i.e., switching time = 2 sec) that we reported in the previous manuscript.

For demonstrating the switching speed of our RRAM under pulsed bias voltages, Figs. 4d-f and the following sentences (now in page 8, from line 17 to line 25) have been added in the revised manuscript.

Figure 4 **d** Circuit diagram of the measurement system for examining the resistive switching speed of the RRAM. **e** Transient responses of the RRAM during the set (1.5 V, 3.8 μs) and reset (-2.0 V, 3.8 μs) processes. Insets: enlarged time trace of the read pulses (0.1 V, 3.8 μs) fed into the RRAM after set (lower left) and reset (upper right) pulse biases. **f** I_{RRAM} vs. pulse width of the applied bias V_{in} for both set (top) and reset (bottom) processes.

For practical applications, a fast switching speed of an RRAM is highly desirable. We characterize the switching speed of our CsPbBr₃ QD-based RRAM using a measurement setup (see Fig. 4d) composed of a pulse generator, an oscilloscope, and a load resistor (50 Ω). The transient switching between the HRS and the LRS, and the corresponding real-time current (I_{RRAM}) are directly monitored by examining the bias change over the load resistor. As shown in Fig. 4e, read pulses (0.1 V, 3.8 μs) are fed into the RRAM before and after the set/reset pulses. The RRAM is switched from the HRS to the LRS when a set pulse (1.5 V, 3.8 μs) is applied, evidenced by the surge of I_{RRAM} in response to the read pulse (inset, lower left). The RRAM is then switched back to the HRS upon applying a reset pulse (-2.0 V, 3.8 μs), and therefore, almost no current is detected by the read pulse (inset, upper right). This fast switching behavior of our RRAM holds even when the pulse widths of the set (1.5 V)/reset (-2.0 V) bias are narrowed down to $\sim 3.0 \mu s$, as can be seen in Fig. 4f.

For demonstrating the fast modulation speed of our LEC, Figs. 3 and the following paragraph (now in page 6 and 7, starting at line 9 from the bottom of page 6) have been added in the revised manuscript.

Figure 3 **a** Circuit diagram illustrating the transient measurement setup for simultaneously monitoring the pulse voltage (V_{in}) applied on the LEC, the electrical current (I_{LEC}) flowing through the LEC, and the optical output-power (P_{LEC}) emitting by the LEC. **b** and **c** Time traces of I_{LEC} and P_{LEC} of the LEC under modulated V_{in} at 10 kHz and 80 kHz. **d** Temporal P_{LEC} in response to a square-wave pulse voltage of V_{in} (magnitude: 6V, pulse width: 50 μ s, and duty cycle: 50%) applied on the LEC. The rise (τ_{Rise}) and fall (τ_{Fall}) times of P_{LEC} are also marked in the figure. **e** The enlarged rising edge of P_{LEC} , where τ_{Delay} , the time delay from the turn-on of V_{in} till the onset of P_{LEC} , is estimated to be $\sim 0.3 \mu$ s. **f** The enlarged falling edge of P_{LEC} with the inset showing the same falling edge but plotted in a semi-logarithmical scale to extract the decay time (τ_{Decay}) of the de-trapped carriers.

In addition to efficient *EL* emissions, a fast response of an LEC is also essential for applications in data communication and signal modulation. Figure 3a schematically depicts the setup for the transient measurement of our CsPbBr₃ QD-based LEC. The light output-power (P_{LEC}) of the LEC modulated by a pulse bias (V_{in}) is detected by a photodiode (PD), and the current (I_{LEC}) flowing through the LEC is analyzed over a series resistor. At the modulation frequency of 10 kHz (Fig. 3b), P_{LEC} and I_{LEC} fully follow the pulse bias applied to the LEC, and both of them are distorted to some extent (especially for P_{LEC} , see Fig. 3c) at a higher modulation frequency of 80 kHz (approaching to the transmission speed regulated for the near-field communication, that is, \sim a hundred of kbit/s). Nevertheless, P_{LEC} is sufficiently saturated and thus can still be read to identify the on/off states of the LEC. Fig. 3d shows temporal response of P_{LEC} in response to a square-wave pulse bias of V_{in} (magnitude: 6V, pulse width: 50 μ s, and duty cycle: 50%), where the response time is defined as the time difference between 10 and 90% of the maximum P_{LEC} . The extracted rise and fall times of P_{LEC} are $\tau_{Rise} \sim 4.9$ and $\tau_{Fall} \sim 5.2 \mu$ s, respectively. Both values are on a microsecond time scale, indicating that the LEC's light emission responds instantly to the pulse bias, and more

importantly the parallel data transmission using LEM is indeed feasible. Fig. 3e plots the enlarged rising edge of P_{LEC} to further examine its onset time τ_{Delay} , the time delay from the initial rise of V_{in} to the onset of P_{LEC}. A τ_{Delay} value of $\sim 0.3 \mu\text{s}$ is extracted, representing the time scale over which the radiative recombination occurs in the perovskites when driven by V_{in}. Given the charge-carrier mobility (μ_{eff}) is expressed as $\mu_{\text{eff}} = L/(\tau_{\text{Delay}} \cdot E)$ ³⁹, where L is the thickness of the active layer and E is the electric field across the LEC, we estimate μ_{eff} of the CsPbBr₃ QDs to be $\sim 1.38 \times 10^{-3} \text{ cm}^2 \cdot \text{V}^{-1} \cdot \text{s}^{-1}$ from the transient P_{LEC} data. The smaller μ_{eff} value as compared to that of bulk perovskites⁴⁰⁻⁴² is expected and results from more scattering boundaries within the QD layer than in the bulk. Upon V_{in} is turned off, the enlarged falling edge of P_{LEC} in Fig. 3f shows that P_{LEC} decays slowly to 3 % in $\sim 8 \mu\text{s}$, followed by a faster decay with a τ_{Decay} value of $\sim 4.5 \mu\text{s}$ (inset, Fig. 3f). The two steps in the falling of P_{LEC} have been observed elsewhere, resulting from the depletion of the carrier reservoir established under pulse bias⁴³, and the radiative recombination of delocalized carriers originally trapped by the defect centers within the perovskites⁴⁴, respectively. We estimate the trap center density to be $1.2 \times 10^{15} \text{ cm}^{-3}$ by integrating P_{LEC} over the entire period of τ_{Decay} , which is in line with other highly-efficient perovskites reported in the literature^{45, 46}.

We have also included new references, and they are now Refs. 39-46 in the revised manuscript.

39. Barth, S. et al. Electron mobility in tris(8-hydroxy-quinoline)aluminum thin films determined via transient electroluminescence from single- and multilayer organic light-emitting diodes. *J. Appl. Phys.* **89**, 3711-3719 (2001).
40. Yu, M., Yi, C., Wang, N., Zhang, L., Zou, R., Tong, Y., Chen, H., Cao, Y., He, Y., Wang, Y., Xu, M., Liu, Y., Jin, Y., Huang, W. & Wang, J. Control of barrier width in perovskite multiple quantum wells for high performance green light-emitting diodes. *Adv. Opt. Mater.* **7**, 1801575 (2019).
41. Herz, L. M. Charge-carrier mobilities in metal halide perovskites: fundamental mechanisms and limits. *ACS Energy Lett.* **2**, 1539-1548 (2017).
42. Motta, C., El-Mellouchi, F. & Sanvito, S. Charge carrier mobility in hybrid halide perovskites. *Sci. Rep.* **5**, 12746 (2015).
43. Lupton, J. M., Nikitenko, V. R., Samuel, I. D. W. & Bäessler, H. Time delayed electroluminescence overshoot in single layer polymer light-emitting diodes due to electrode luminescence quenching. *J. Appl. Phys.* **89**, 311 (2001).
44. Cheon, K. O. & Shinar, J. Electroluminescence spikes, turn-off dynamics, and charge traps in organic light-emitting devices. *Phys. Rev. B* **69**, 201306(R) (2004).
45. Xu, M., Peng, Q., Zou, W., Gu, L., Xu, L., Cheng, L., He, Y., Yang, M., Wang, N., Huang, W. & Wang, J. A transient-electroluminescence study on perovskite light-emitting diodes. *Appl. Phys. Lett.* **115**, 041102 (2019).

46. Xing, G., Wu, B., Wu, X., Li, M., Du, B., Wei, Q., Guo, J., Yeow, E. K. L., Sum, T. C. & Huang, W. Transcending the slow bimolecular recombination in lead-halide perovskites for electroluminescence. *Nature. Comm.* **8**, 14558 (2017).

5. The investigation about this LEM system is still relatively simple. Can authors propose several specific future potential applications of this system?

Response: We have significantly broadened the scope of our investigation of the all-perovskite LEM system in terms of device performance and application. *For the former*, first we have shown that for a single Ag/CsPbBr₃ QDs/ITO device, its functionality can be swiftly switched between the RRAM and the LEC operations as fast as 1 kHz by simply modulating its bias polarity (Fig. 5). It is this fast switching capability of our single perovskite device that increases the functionality of the all-perovskite LEM. We have then tested the speed of the all-perovskite LEM, demonstrating that the device can operate as fast as 5 kHz under both positive and negative pulsed biases (Fig. 7 a and b). We have further demonstrated that the RRAM and the LEC in an all-perovskite LEM device can be actively switched at 1 kHz, much faster than that (i.e., switching time = 2 sec) reported in the previous manuscript. Finally, we have employed QDs with two different sizes, one emitting at $\lambda_1 = 532$ nm and the other at $\lambda_2 = 515$ nm, realizing a two-color emitting LEM device (Fig. 7c and d). *For the latter*, our fast, all-perovskite LEM promises parallel electrical and optical readings of information, in which the data stored in the LEM can be *electrically* read by the RRAM and *optically* transmitted through the light emission from the LEC. The two opposite situations of the LEM realized by simply changing the bias polarity could represent two distinct states in the logic operation, and hence could be further utilized to enable the desirable multilevel data storage. Moreover, the dual light-emitting and resistive-switching operations of the LEM with fast response could potentially implement a novel scheme for data encryption, where the light emission from the LEC is transmitted and detected by an optical sensor, and then the detected signal is converted into the encryption key for the user to access the encoded information recorded in the RRAM. Lastly, the two-color emitting LEM enables a real-time reading of its digital status (i.e., write or erase) simply by the emission colors. We foresee that based on the present demonstration on the fast operation and active switching of the all-perovskite LEM, more application scenarios are possible by further increasing the device functionalities.

We have included in the revised manuscript the experimental data on the performance (including speed, switching, and two emission colors) of our all-perovskite LEM in Fig. 7, along with two new paragraphs now in page 11-13, starting at line 8 from the bottom of page 11.

Figure 7 **a** and **b** Time traces of V_{RRAM} and P_{LEC} of the all-perovskite LEM under both forward (+ 6 V, left panel) and reverse (- 6 V, right panel) pulse bias V_{in} (100 μ s width with 50% duty cycle) at 5 kHz, demonstrating the fast, parallel optical and electrical readings of the LEM regardless of the bias polarity. The equivalent circuit diagrams indicating the functionality of each device in the LEM under both bias polarities are also inserted in the top panels. **c** EL spectra of the two-color emitting all-perovskite LEM employing CsPbBr₃ QDs with two different sizes, one emitting at $\lambda_1 = 532$ nm (exhibiting green color, see the photograph inset in the top panel) while the other emitting at $\lambda_2 = 515$ nm (appearing aqua luster, see the photograph inset in the bottom panel). The corresponding circuit diagrams of the LEM when it appears green (top panel) or aqua (bottom panel) are also inserted. These two colors could serve as a real-time indicator of the digital status (i.e., write or erase) of the LEM. **d** Time traces of the current (I) and the emitting optical power (P_{LEC}) of the two-color emitting LEM when actively switching the V_{in} polarity at 1 kHz. The optical difference in the P_{LEC} levels of the two emission colors (bottom panel) are employed in this work to distinguish in real time the write and the erase state of the LEM.

Finally, we test the operation speed of our all-perovskite LEM by applying a pulse voltage bias (pulse width 100 μ s with 50% duty cycle), and reading the voltage response of the RRAM (V_{RRAM}) and the light emission of the LEC (P_{LEC}) in the LEM. As indicated by the circuit diagram in the top panel of the Fig. 7a, when the applied pulse voltage (V_{in}) is 6 V (that is, the LEM operates in the region (IV) of the Fig. 6), the negatively biased left device (denoted as *Device 1*) of the LEM behaves as an LEC, while the positively biased device on the right (denoted as *Device 2*) acts as an RRAM. Clearly, under this forward bias operation, both the V_{RRAM} (middle panel) and the P_{LEC}

(bottom panel) traces keep up well with the V_{in} , demonstrating that our LEM can operate as fast as 5 kHz, at which the data stored in the LEM can be *electrically* read by the *Device 2* and *optically* transmitted through the parallel light emission from the *Device 1*. As pointed out earlier that the speed of individual CsPbBr₃ QD-based LEC and RRAM depends strongly on the ion generation and movement in the perovskite QD active layer, the operation speed of our all-perovskite LEM could potentially be enhanced by improving the intrinsic transport properties of the perovskites^{49,50}. The fast operation of our LEM remains if the polarity of the pulse voltage is reversed (see Fig. 7b), except for the fact that the *Device 1* now becomes an RRAM with a negative voltage response and *Device 2* functions as an LEC with a similar light emission power. Note that these two opposite situations of the LEM realized by simply changing the applied voltage polarity could represent two distinct states in the logic operation, and hence could be further utilized to enable the desirable multilevel data storage. Also note that such dual light-emitting and resistive-switching operations of the LEM with fast response could potentially implement a novel scheme for data encryption, where the light emission from the LEC is transmitted and detected by an optical sensor, and then the detected signal is converted into the encryption key for the user to access the encoded information recorded in the RRAM.

In addition to providing a parallel, non-contact optical reading, the emission color of the LEC can be employed as a real-time indicator of the LEM status (i.e., either in the “write” or the “erase” state). To this end, we experimentally realize a two-color emitting LEM by reducing the size of the perovskite QDs in the *Device 2* to only ~ 20 nm for blue-shifting its emission wavelength due to the quantum confinement effect (Supplementary Fig. 2), while keeping the dots in the *Device 1* the same (as those in the Fig. 1c). Under the forward ac operation (+ 6 V with a 250 μ s pulse width), the *Device 2* in the LEM plays the role of an ON-state RRAM, whereas the *Device 1* operates as an emitting LEC with the emission wavelength of $\lambda_1 = 532$ nm (top panel in Fig. 7c). The write state of the LEM can thus be visually recognized by the greenish appearance of the *Device 1*. Upon reversing the ac bias voltage, the LEM turns to its erase mode, in which the *Device 2* originally as an RRAM is switched off and becomes an LEC exhibiting the aqua luster, resulting from the emission of the smaller dots at $\lambda_2 = 515$ nm (bottom panel in Fig. 7c). As a result, one can “perceive” in real time (at the speed of 1 kHz) the electrically encoded digital status of the LEM, by using an optical filter to block one emission color and transmit the other, or using two optical detectors to separately register them from the LEM. Note that in the present proof-of-concept work, these two schemes are not experimentally pursued due to the limitation of our experimental setup, and we instead distinguish two colors of the LEM by differentiating their P_{LEC} levels in a single detector (Fig. 7d). It is worth noting that this real-time

recognition of the digital status of the LEM by simply observing its emission color is enabled by the fast switching of the single perovskite device, as demonstrated earlier in Fig. 5. Although the switching speed between two perovskite devices in the LEM is still slower than that of the state-of-the-art RRAM for performing data processing in series^{51,52}, we expect that it could be further improved by optimizing our perovskite synthesis and the perovskite device structure (such as device dimension, geometry, and layer thickness). Apart from the optimization on the device level, by employing the LEM in a more complicated network structure such as multicast mesh network^{53,54}, we envision that the overall transmitted data rate of the LEM could be further enhanced. In any case, the present demonstration on the fast operation and active switching of the all-perovskite LEM successfully sets a new benchmark for the development of more advanced LEM technologies. On a broader scale, this work also provides a new paradigm for generating novel device concepts and functionalities by employing the synergistic combination of electronic and photonic degrees of freedom in a single material system.

We have also included new references, and they are now Refs. 49-54 in the revised manuscript.

49. Begum, R., Parida, M. R., Abdelhady, A. L., Murali, B., Alyami, N. M., Ahmed, G. H., Hedhili, M. N., Bakr, O. M. & Mohammed, O. F. Engineering interfacial charge transfer in CsPbBr₃ perovskite nanocrystals by heterovalent doping. *J. Am. Chem. Soc.*, **139**, 731–737 (2017).
50. Chen, Z., Brocks, G, Tao, S. & Bobbert, P. A. Unified theory for light-induced halide segregation in mixed halide perovskites. *Nature. Comm.* **12**, 2687 (2021).
51. Wu, Q., Banerjee, W., Cao, J., Ji, Z., Li, L. & Liu, M. Improvement of durability and switching speed by incorporating nanocrystals in the HfO_x based resistive random access memory devices. *Appl. Phys. Lett.* **113**, 023105 (2018).
52. Menzel, S., Witzleben, M. V., Havel, V. & Böttger, U. The ultimate switching speed limit of redox-based resistive switching devices. *Faraday Discuss.*, **213**, 197-213 (2019).
53. Smadi, M. N., Ghosh, S. C, Farid, A. A. Todd, T. D., Hranilovic, S. Free-space optical gateway placement in hybrid wireless mesh networks. *J. Light. Technol.* **27**, 2688-2697 (2009).
54. Abadi, M. M., Cox, M. A., Alsaigh, R. E., Viola, S., Forbes, A. & Lavery, M. P. J. A space division multiplexed free-space-optical communication system that can auto-locate and fully self align with a remote transceiver. *Sci. Rep.* **9**, 19687 (2019).

■ [Response to the comments of Reviewer 3]

This communication reports that all-inorganic CsPbBr₃ perovskite quantum dots (QDs) based Ag/CsPbBr₃/ITO device exhibited the characteristics of light-emitting as well as electrical memory characteristics using electrical field-induced ionic motions. The fabricated devices could act as all-perovskite light emitting memory device for simultaneous electronic and optical reading of the encoded information in communication and computation applications. The communication demonstrates a novel device concept, which could drive the new application of high performance perovskite quantum dots based optoelectronic devices. Here I have the following comments for the authors to address before further considering the acceptance of the manuscript:

1. In page 4, the authors described “The transfer of Cs⁺, Pb²⁺, and Br⁻ ions from the soluble to insoluble solvents is too fast during the supersaturated recrystallization process, which significantly enlarges the QD size to ~ 50 –100 nm, larger than the CsPbBr₃ QDs synthesized by the hot-injection method that generally have a smaller size of ~ 10 –15 nm”. The size of perovskite QD significantly affected the electronic and optoelectronic characteristics and the long-term stability of the fabricated devices. Here I suggest the authors to discuss the device performance using different sizes of the QD.

Response: We thank the Reviewer’s valuable suggestion. The size-tunable *PL* emissions of the CsPbBr₃ QDs enabled by the quantum confinement effect (see Supplementary Figure 2) offer a great opportunity to significantly enhance the all-perovskite LEM device functionalities. We have employed QDs with two different sizes, one emitting at $\lambda_1 = 532$ nm and the other at $\lambda_2 = 515$ nm, realizing a two-color emitting LEM device (Fig. 7c and d). The resulting LEM allows for the real-time reading of its digital status (i.e., write or erase state) simply by the emission colors. This demonstration significantly broadens the scope of applications of the proposed all-perovskite LEM.

For demonstrating the tunable bandgaps of the CsPbBr₃ resulting from the quantum confinement effect and their importance in the CsPbBr₃ QD-based LEM functionalities, we have included the relevant data in Supplementary Figure 2 in the Supplementary Information of the revised manuscript.

Supplementary Figure 2 | Quantum confinement effect of the CsPbBr₃ QDs

Supplementary Figure 2 a Top-view SEM images of the CsPbBr₃ QDs with different sizes obtained by varying the *OA*: *OAm* ratios of the stabilization solution during the synthesis. The size distributions of the synthesized dots (in percentage) are also inserted in the images. The scale bars are 200 nm in all SEMs. **b** *PL* spectra of the synthesized CsPbBr₃ QDs with different sizes. **c** Size dependence of the CsPbBr₃ QDs on the peak emission wavelength of their *PL* spectra.

To explore the tunable bandgaps of the CsPbBr₃ resulting from the quantum confinement effect and their importance in the CsPbBr₃ QD-based LEM functionalities, we systematically reduce the size of the CsPbBr₃ QDs by changing the mixture ratios between the Oleic acid (*OA*) and the oleylamine (*OAm*) during the stabilization process for the PbBr₂/CsBr precursor solution. The average sizes of the synthesized CsPbBr₃ QDs (all with the same stoichiometry) are 123, 63, and 20 nm, for the *OA*: *OAm* ratios of 0.5 ml: 0.25 ml, 1.0 ml: 1.0 ml, and 1.5 ml:1.5 ml, respectively (Supplementary Figure 2a). The size distributions of the synthesized CsPbBr₃ QDs (in percentage) are also estimated and inserted in the figures. Clearly, gradually reducing the QD size blueshifts the peak emission wavelength of the *PL* spectra, from 526 to 520, and then 511 nm, due to the increased quantum confinement effect (Supplementary Figure 2b and 2c). The demonstrated tunable bandgaps of the CsPbBr₃ QDs offer a great opportunity to significantly enhance the LEM device functionalities, as evidenced by the realization of the two-color emitting LEM device employing two different QDs (see Figure 7 in the main text).

We have also included Fig. 7c and d, and a paragraph (now in page 12 and 13, starting at line 11 in page 12) in the revised manuscript for the demonstration of the two-color emitting LEM.

Figure 7 **c** EL spectra of the two-color emitting all-perovskite LEM employing CsPbBr₃ QDs with two different sizes, one emitting at $\lambda_1 = 532$ nm (exhibiting green color, see the photograph inset in the top panel) while the other emitting at $\lambda_2 = 515$ nm (appearing aqua luster, see the photograph inset in the bottom panel). The corresponding circuit diagrams of the LEM when it appears green (top panel) or aqua (bottom panel) are also inserted. These two colors could serve as a real-time indicator of the digital status (i.e., write or erase) of the LEM. **d** Time traces of the current (I) and the emitting optical power (P_{LEC}) of the two-color emitting LEM when actively switching the V_{in} polarity at 1 kHz. The difference in the P_{LEC} levels of the two emission colors (bottom panel) are employed in this work to distinguish in real time the write and the erase state of the LEM.

In addition to providing a parallel, non-contact optical reading, the emission color of the LEC can be employed as a real-time indicator of the LEM status (i.e., either in the “write” or the “erase” state). To this end, we experimentally realize a two-color emitting LEM by reducing the size of the perovskite QDs in the *Device 2* to only ~ 20 nm for blue-shifting its emission wavelength due to the quantum confinement effect (Supplementary Fig. 2), while keeping the dots in the *Device 1* the same (as those in the Fig. 1c). Under the forward ac operation (+ 6 V with a 250 μ s pulse width), the *Device 2* in the LEM plays the role of an ON-state RRAM, whereas the *Device 1* operates as an emitting LEC with the emission wavelength of $\lambda_1 = 532$ nm (top panel in Fig. 7c). The write state of the LEM can thus be visually recognized by the greenish appearance of the *Device 1*. Upon reversing the ac bias voltage, the LEM turns to its erase mode, in which the *Device 2* originally as an RRAM is switched off and becomes an LEC exhibiting the aqua luster, resulting from the emission of the smaller dots at $\lambda_2 = 515$ nm (bottom panel in Fig. 7c). As a result, one can “perceive” in real time (at the speed of 1 kHz) the electrically encoded digital status of the LEM, by using an optical filter to block one emission color and transmit the other, or using two optical detectors to separately register them from the LEM. Note that in the present proof-of-concept work, these two schemes are not experimentally pursued due to the limitation of our

experimental setup, and we instead distinguish two colors of the LEM by differentiating their P_{LEC} levels in a single detector (Fig. 7d). It is worth noting that this real-time recognition of the digital status of the LEM by simply observing its emission color is enabled by the fast switching of the single perovskite device, as demonstrated earlier in Fig. 5. Although the switching speed between two perovskite devices in the LEM is still slower than that of the state-of-the-art RRAM for performing data processing in series^{51,52}, we expect that it could be further improved by optimizing our perovskite synthesis and the perovskite device structure (such as device dimension, geometry, and layer thickness). Apart from the optimization on the device level, by employing the LEM in a more complicated network structure such as multicast mesh network^{53,54}, we envision that the overall transmitted data rate of the LEM could be further enhanced. In any case, the present demonstration on the fast operation and active switching of the all-perovskite LEM successfully sets a new benchmark for the development of more advanced LEM technologies. On a broader scale, this work also provides a new paradigm for generating novel device concepts and functionalities by employing the synergistic combination of electronic and photonic degrees of freedom in a single material system.

In addition, a couple of new references (now Refs. 51-54 in the revised manuscript) are included.

51. Wu, Q., Banerjee, W., Cao, J., Ji, Z., Li, L. & Liu, M. Improvement of durability and switching speed by incorporating nanocrystals in the HfO_x based resistive random access memory devices. *Appl. Phys. Lett.* **113**, 023105 (2018).
 52. Menzel, S., Witzleben, M. V., Havel, V. & Böttger, U. The ultimate switching speed limit of redox-based resistive switching devices. *Faraday Discuss.*, **213**, 197-213 (2019).
 53. Smadi, M. N., Ghosh, S. C., Farid, A. A., Todd, T. D., Hranilovic, S. Free-space optical gateway placement in hybrid wireless mesh networks. *J. Light. Technol.* **27**, 2688-2697 (2009).
 54. Abadi, M. M., Cox, M. A., Alsaigh, R. E., Viola, S., Forbes, A. & Lavery, M. P. J. A space division multiplexed free-space-optical communication system that can auto-locate and fully self align with a remote transceiver. *Sci. Rep.* **9**, 19687 (2019).
2. The authors used poly(methyl methacrylate) (PMMA) as the protection layer on top of the CsPbBr₃ QDs. Why does it need the protection layer? Also, what is the criteria on the selection of the materials for the protection layer?

Response: The PMMA protection layer is needed to prevent the Ag penetration during the sputtering deposition, as evidenced by Supplementary Figure 3, which in turn minimizes the formation of current leakage paths between the top (Ag) and the bottom (ITO) electrodes, and enhances the performance and the stability of the fabricated LEM. The reasons that we use PMMA as the protection layer in our device are two folds. First,

it can be prepared by a simple spin-coating process with minimal heating, which is highly compatible with the underlying CsPbBr₃ QD layer. Moreover, the PMMA can also improve the surface roughness of the QD layer (Supplementary Figure 4), which facilitates the formation of the top Ag electrode. Note that the electrically non-conducting PMMA layer cannot be too thick (~ 25 nm thick in our work) otherwise it would significantly degrade the electrical properties of the RRAM device. Also note that the incorporation of PMMA in the CsPbBr₃ QD-based RRAM to enhance its device stability has been demonstrated previously (*Adv. Mater.* **30**(28), 1800327 (2018)), so we employ the same strategy in our work.

3. In figure 3d, the retention characteristic decreased from 10^3 to 10^2 after 5000 seconds of operation. This is far below the standard compared to the reported resistive memory devices. I suggest the authors to improve the retention characteristics for the potential practical applications.

Response: We have significantly improved the retention of our RRAM device. As can be seen in Supplementary Figure 7a, both the HRS and the LRS (read at 0.1 V) of the RRAM can retain for over 10^5 sec (that is, much longer than 5000 sec reported in the previous manuscript) and the ON/OFF current ratio keeps at $\sim 10^2$, indicating good retention stability. We also conduct an endurance test on our RRAM under a pulse sweeping mode (Supplementary Fig. 7b), and except for some fluctuations, the HRS/LRS ratio remains basically the same even after more than 10^4 sweeping cycles. Moreover, the RRAM can be switched reversibly between the HRS and the LRS more than 50 times when subject to repeated sweeping processes between a positive and a negative dc bias (Supplementary Fig. 7c), and its *I-V* characteristic shows a small device-to-device variation (Supplementary Fig. 8). All of these validate the reliable and reproducible write/erase characteristics of our CsPbBr₃ QD-based RRAM.

For proving the reliability performance of our RRAM, the following sentences (now in page 8, starting at line 25 all the way to the top of page 9) along with Supplementary Figures 7 and 8 and their descriptions have been added in the revised manuscript.

As far as the device reliability is concerned, both the HRS and the LRS (read at 0.1 V) of the RRAM can retain for over 10^5 sec and the ON/OFF current ratio keeps at $\sim 10^2$ (Supplementary Fig. 7a), indicating good retention stability. We also conduct an endurance test on our RRAM under a pulse sweeping mode (Supplementary Fig. 7b), and except for some fluctuations, the HRS/LRS ratio remains basically the same even after more than 10^4 sweeping cycles. Moreover, the RRAM can be switched reversibly between the HRS and the LRS more than 50 times when subject to repeated sweeping processes between a positive and a negative dc bias (Supplementary Fig. 7c), and its *I-V* characteristic shows a small device-to-device variation (Supplementary Fig. 8). All of

these validate the reliable and reproducible write/erase characteristics of our CsPbBr₃ QD-based RRAM.

Supplementary Figure 7 | Retention and endurance tests of the RRAM mode of the CsPbBr₃ QD-based LEM

Supplementary Figure 7 a Retention performance of LRS and HRS of an Ag/CsPbBr₃ QDs/ITO device when acting as an RRAM. b Endurance test result (read at 0.1 V) of the device over 10⁴ cycles. c *I-V* characteristics of the device after different numbers of times of a dc bias sweep (0 V → +2 V → 0 V → -2 V → 0 V) up to 50 times.

Supplementary Figure 7a shows the retention performance of the RRAM mode of a single Ag/CsPbBr₃ QDs/ITO device by consecutively applying square-wave bias pulses (0.1 V on a 1.1 sec period) to read the RRAM. Both HRS and LRS can retain for over 10⁵ sec and the ON/OFF current ratio keeps at ~ 10², indicating good retention stability. Supplementary Figure 7b shows the endurance test results of the device under a modulated bias with a repetitive sequence of 1.5 V(Set)/ 0.1 V(Read)/ - 2.0 V(Reset)/0.1 V(Read). Except for some fluctuations in both HRS and LRS, there is no obvious decline of HRS/LRS ratio even after more than 10⁴ sweeping cycles. Moreover, as shown in Supplementary Figure 7c, the RRAM can be switched reversibly between its HRS and LRS more than 50 times by repeatedly sweeping a dc bias in a sequence of 0 V → +2 V → 0 V → -2 V → 0 V. The current ratio between HRS and LRS could maintain at ~ 10² during these alternating positive/negative dc bias sweeps. These results altogether ensure the reliable and reproducible write/erase characteristics of either one of the Ag/CsPbBr₃ QDs/ITO devices in our CsPbBr₃ QD-based LEM when acting as an RRAM.

Supplementary Figure 8 | Device-to-device variation in the set and reset voltages of the CsPbBr₃ QD-based RRAM

Supplementary Figure 8 a *I-V* characteristics of 18 CsPbBr₃ QD-based RRAM devices by setting the compliance current to $I_{CC} = 1$ mA. **b** Cumulative probability distributions of the set (right) and reset (left) voltages.

Supplementary Figure 8a shows the *I-V* characteristics of total 18 CsPbBr₃ QD-based RRAM devices to study the device-to-device variability in their set and reset voltages. Supplementary Figure 8b plots the cumulative probability distribution of the set and reset voltages extracted from Supplementary Figure 8a. It is clear that the set process of the RRAMs occurs mostly at the mean value of $\mu = 0.66$ V with a small standard deviation of $\sigma = 0.04$, though the reset process shows a slightly larger variability ($\sigma = 0.105$) from the mean value of $\mu = -1.35$ V. The above observations validate that our CsPbBr₃ QD-based RRAM exhibits a good device-to-device variability.

4. The authors showed green color light-emission in the fabricated devices. I suggest the authors to show the multi-color emission and demonstrate that the methodology could be applied to other kind of perovskite QD based devices.

Response: As already stated in in our response to the Reviewer’s question #1, we have demonstrated an all-perovskite LEM emitting green and aqua. Other emission colors such as blue and red are certainly possible by employing different perovskite QDs

Thank you again for your consideration of the manuscript. We sincerely hope that the revised manuscript can be accepted for publication in Nature communications.

Sincerely yours,
Ya-Ju Lee, Ph.D.
Professor

REVIEWERS' COMMENTS

Reviewer #1 (Remarks to the Author):

The authors have responded to the comments with carrying out new experiments and presenting detailed explanation. The switching speed of the device, cycling and retention behavior were measured. Although I am not sure about the practical application of LEM, of which performance is less than the counterparts RAM and LED, I am happy to see the authors' efforts to revise the manuscript considerably. I do recommend the publication of the manuscript in the journal at the current format.

Reviewer #2 (Remarks to the Author):

The authors addressed all my comments in detail. I recommend accepting the paper.

RESPONSE TO REVIEWERS' COMMENTS

■ [Response to the comments of Reviewer 1]

1. The authors have responded to the comments with carrying out new experiments and presenting detailed explanation. The switching speed of the device, cycling and retention behavior were measured. Although I am not sure about the practical application of LEM, of which performance is less than the counterparts RAM and LED, I am happy to see the authors' efforts to revise the manuscript considerably. I do recommend the publication of the manuscript in the journal at the current format.

Response: We thank the viewer's comments and the recommendation of our manuscript for publication in Nature Communications.

■ [Response to the comments of Reviewer 2]

The authors addressed all my comments in detail. I recommend accepting the paper.

Response: We thank the viewer's comments and the recommendation of our manuscript for publication in Nature Communications.

We thank the reviewers again for their support.

Sincerely yours,
Ya-Ju Lee, Ph.D.
Professor